# R-spondins are BMP receptor antagonists in *Xenopus* early embryonic development

Hyeyoon Lee[1], Carina Seidl [1], Rui Sun[1], Andrey Glinka[1] & Christof Niehrs[1,2]✉

BMP signaling plays key roles in development, stem cells, adult tissue homeostasis, and disease. How BMP receptors are extracellularly modulated and in which physiological context, is therefore of prime importance. R-spondins (RSPOs) are a small family of secreted proteins that co-activate WNT signaling and function as potent stem cell effectors and oncogenes. Evidence is mounting that RSPOs act WNT-independently but how and in which physiological processes remains enigmatic. Here we show that RSPO2 and RSPO3 also act as BMP antagonists. RSPO2 is a high affinity ligand for the type I BMP receptor BMPR1A/ALK3, and it engages ZNRF3 to trigger internalization and degradation of BMPR1A. In early *Xenopus* embryos, Rspo2 is a negative feedback inhibitor in the BMP4 synexpression group and regulates dorsoventral axis formation. We conclude that R-spondins are bifunctional ligands, which activate WNT- and inhibit BMP signaling via ZNRF3, with implications for development and cancer.

[1] Division of Molecular Embryology, DKFZ-ZMBH Alliance, Deutsches Krebsforschungszentrum (DKFZ), 69120 Heidelberg, Germany. [2] Institute of Molecular Biology (IMB), 55128 Mainz, Germany. ✉email: niehrs@dkfz-heidelberg.de

Bone Morphogenetic Proteins (BMPs) are a subfamily of TGFβ growth factors that exert a plethora of crucial functions in embryonic development, adult tissue homeostasis, as well as regeneration, and they underlie human pathology such as skeletal disorders, cancer, and fibrosis in multiple organs[1–5]. Due to their accessibility, extracellular components of the BMP pathway are of particular interest as therapeutic targets[6] and mechanistic understanding of receptor modulation should improve the ability to manipulate BMP-dependent processes.

BMPs signal through a tetrameric receptor kinase complex composed of type I (BMPR1A/ALK3, BMPR1B/ALK6, ACVR1/ALK2, or ACVRL1/ALK1) and type II receptors (BMPR2, ACVR2A, ACVR2B)[7]. Ligands and receptors combine in a combinatorial fashion[8] and phosphorylate SMAD1, 5, and 8, which enter the nucleus with SMAD4 to regulate target gene expression[9,10]. There exists a multitude of extracellular modulators of TGFβ signaling, either soluble or membrane-associated proteins that control ligand availability, processing, ligand–receptor interaction, and receptor activation[11]. However, only two BMP receptor antagonists are known, which directly bind and inhibit receptor function, the TGFβ-family proteins BMP3 and Inhibin[12,13].

R-spondins (RSPO1-4) are a family of four secreted ~30 kDa proteins implicated in development and cancer[14–20]. RSPOs are a key ingredient to maintain organoid cultures where they stimulate stem cell growth[21,22]. They amplify WNT signaling by preventing Frizzled/LRP5/6 receptor ubiquitination and degradation via transmembrane E3 ubiquitin ligases ring finger 43 (RNF43) and zinc and ring finger 3 (ZNRF3), thereby sensitizing cells to WNT ligands[14,23–25]. RSPOs bind to ZNRF3/RNF43 and to the stem cell marker Leucine-rich repeat containing G protein-coupled receptor 5 (LGR5), and two related proteins, LGR4 and LGR6, leading to the internalization of the RSPO-LGR-ZNRF3/RNF43 complex and lysosomal degradation[14,17,26]. RSPOs harbor two furin-like repeats (FU1, FU2) domains that bind to ZNRF3/RNF43 and LGRs, respectively[27]. In addition, they contain a thrombospondin 1 (TSP1) domain, which possess about 40% overall sequence homology[24,28]. The TSP1 domain is not essential for WNT/LRP6 signaling but it binds to HSPGs (Heparan Sulfate Proteoglycans) and thereby promotes WNT5A/PCP (planar cell polarity) signaling[24,29].

Unexpectedly, recent studies showed that RSPO2 and RSPO3 can potentiate WNT signaling in the absence of all three LGRs in vitro and in vivo[27,30]. Moreover, WNT and RSPO ligands are functionally non-equivalent since e.g., WNT ligand overexpression cannot induce crypt expansion in contrast to RSPO2 or RSPO3[31] and RSPO2 and WNT1 have distinct effects on mammary epithelial cell growth[32] and cochlea development[33]. Hence, these inconsistencies in our current understanding raise the questions: do RSPOs possess WNT-independent functions? Do they engage other receptors? If so, in which physiological processes is this relevant?

Here we show that RSPO2 and RSPO3 are high affinity ligands for the BMP receptor BMPR1A/ALK3. RSPO2 forms a ternary complex between BMPR1A and the E3 ligase ZNRF3, which triggers endocytosis and degradation of the BMP receptor. We show that Rspo2 antagonizes BMP signaling during embryonic axis formation in Xenopus. By gain-of-function and loss-of-function experiments rspo2 cooperates with Spemann organizer effectors to regulate the BMP morphogen gradient, which controls dorsoventral axis formation. Our study reveals R-spondins as BMP receptor antagonists in development, inviting re-interpretation of the mode of action of R-spondins and ZNRF3 in stem cell and cancer biology.

## Results

### RSPO2 and RSPO3 antagonize BMP4 signaling independently of WNT.

In considering possible WNT-independent functions of RSPOs, we revisited our early observation that rspo2 overexpression affected BMP signaling in Xenopus embryos[20]. We tested if RSPO2 could suppress BMP signaling in human cells. To this end, we utilized human hepatocellular carcinoma (HEPG2) cells, which express very low levels of RSPOs (Supplementary Fig. 1a). Intriguingly, treatment with RSPO2 and RSPO3 but not RSPO1 and RSPO4 decreased BMP4 signaling, while all RSPOs showed similar ability to amplify WNT signaling (Fig. 1a, Supplementary Fig. 1b). Importantly, inhibition of BMP signaling by RSPO2 and RSPO3 was independent of WNT/β-catenin signaling, since it remained unaffected by siRNA knockdown of β-catenin (Fig. 1b, Supplementary Fig. 1c, d). RSPO2 and RSPO3, but not RSPO1 and RSPO4 treatment decreased phosphorylation of Smad1, which is a hallmark of BMP signaling activation (Fig. 1c, d, Supplementary Fig. 1e, f). Focusing on RSPO2, we confirmed that RSPO2 overexpression decreased Smad1 phosphorylation and treatment with RSPO2 protein decreased BMP target ID1 expression (Supplementary Fig. 1g, Fig. 1e). Inhibition of BMP signaling by RSPO2 was unaffected by siRNA knockdown of LGR4/5, LRP5/6, DVL1/2/3, and ROR1/2 (Fig. 1f, g, Supplementary Fig. 1h–j), suggesting independence of WNT/LRP and WNT/PCP signaling. Moreover, different from RSPO2, treatment with WNT3A, WNT3A surrogate[34], or the WNT antagonist DKK1 had no effect on BMP signaling (Fig. 1h, Supplementary Fig. 1k, l), corroborating WNT-independent RSPO2 function.

To delineate the domains required for BMP inhibition, we analyzed deletion mutants of RSPO2 and found both the TSP1-domains and FU-domains to be important for signaling inhibition (Fig. 1i, j)[24]. We next investigated RSPO2 deficiency in H1581 cells, a human large cell lung carcinoma cell line that expresses high levels of RSPO2 (Supplementary Fig. 1a). Knockdown of RSPO2 but not LRP5/6 sensitized H1581 cells to BMP stimulation (Fig. 1k, l, Supplementary Fig. 1m, n). We conclude that RSPO2 and RSPO3 antagonize BMP signaling independently of WNT signaling.

### Rspo2 antagonizes BMP signaling during Xenopus embryonic axis development.

To analyze if Rspo2 inhibits BMP signaling in vivo, we turned to early Xenopus development. In the early amphibian embryo, the Spemann organizer is a small evolutionary conserved signaling center, which plays an eminent role in regulating embryonic axis formation and neural induction. One essential molecular mechanism underlying Spemann organizer function resides in its secretion of BMP antagonists, which create a BMP morphogen gradient that patterns the embryo[35–37]. Since rspo2 is expressed and functions in WNT-mediated myogenesis of early Xenopus embryos[20], we analyzed if it may have an additional role as BMP antagonist in axial patterning.

bmp4 overexpression ventralizes Xenopus embryos, resulting in small heads and enlarged ventral structures[38]. Injection of wild-type rspo2 mRNA, but neither its ΔFU1/2 nor ΔTSP1 deletion mutants rescued these bmp4-induced malformations (Fig. 2a, b). This domain requirement is different from that for WNT signaling activation, where only FU1 and FU2 but not the TSP1 domain are essential[20]. Conversely, injection of a previously characterized rspo2 antisense Morpholino (Mo)[20] increased endogenous BMP signaling, and this was unaffected by lrp6 Mo (Fig. 2c)[39]. Strikingly, coinjection of bmp4 Mo and rspo2 Mo neutralized each other in BMP signaling reporter assay (Fig. 2d), BMP target gene expression (vent1, sizzled) (Fig. 2e, f), as well as defects in dorsoventral axis development (Supplementary Fig. 2a, b). Typically, overexpression of common BMP antagonists such as noggin or chordin that sequester BMP ligands, leads to strongly dosalized Xenopus embryos, with enlarged heads and cement

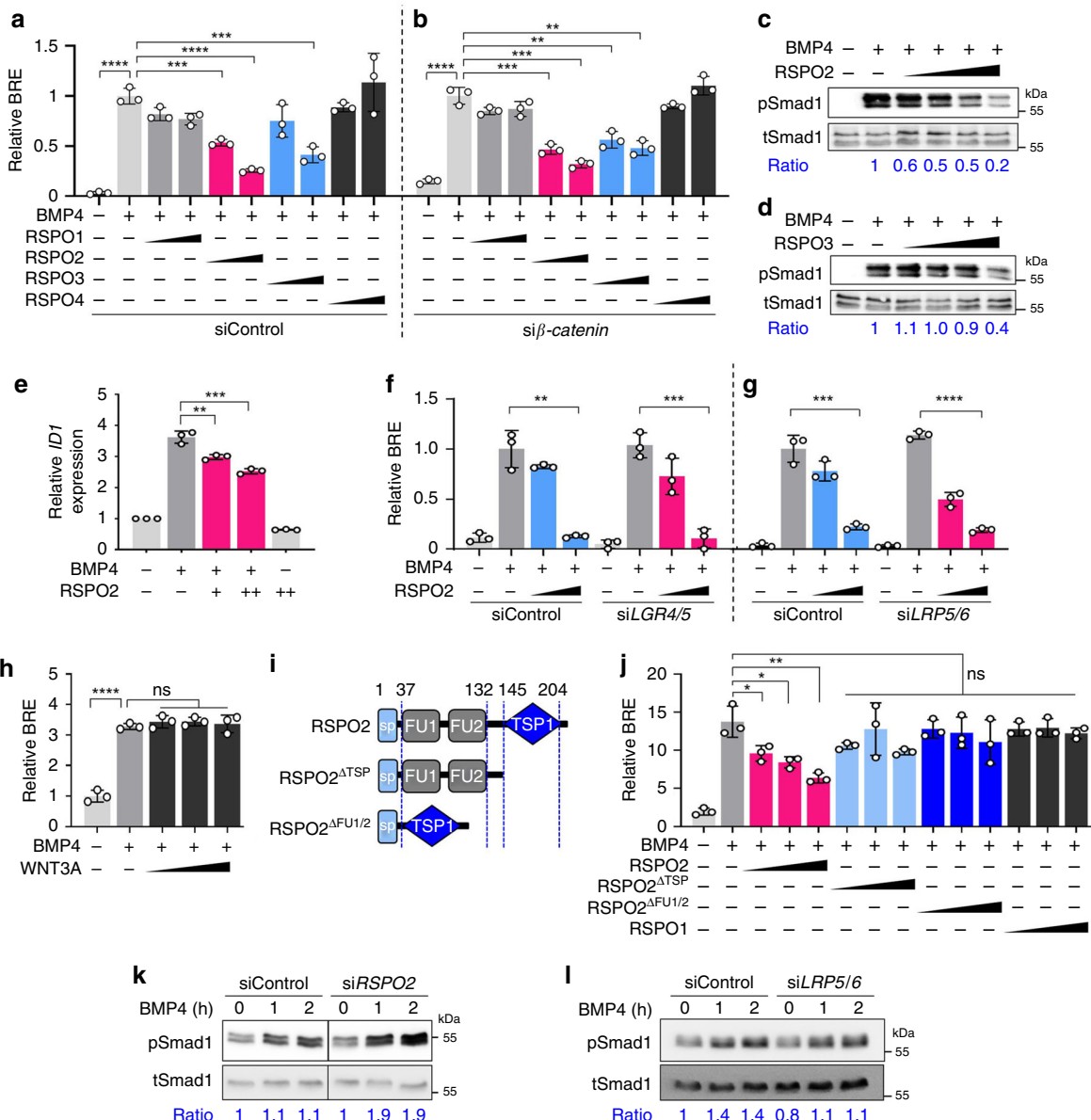

**Fig. 1 RSPO2 and RSPO3 antagonize BMP4 signaling WNT independently. a, b** BRE reporter assay in HEPG2 cells upon siControl (**a**) or siβ-catenin (**b**) transfection, with or without overnight BMP4 and RSPO1-4 treatment as indicated. n = 3 biologically independent samples. **c, d** Western blot analyses of phosphorylated Smad1 (pSmad1) and total Smad1 (tSmad1) in HEPG2 cells stimulated by BMP4, treated with or without increasing amount of RSPO2 (**c**) or RSPO3 (**d**) overnight. Cells were starved 3–6 h before the stimulation. Ratio, relative levels of pSmad1 normalized to tSmad1. Representative data from two independent experiments are shown. **e** qRT-PCR analysis of BMP target ID1 in HEPG2 cells upon BMP4, with or without overnightRSPO2 treatment. n = 3 experimentally independent samples. Data are displayed as means ± SD. **P < 0.01, ***P < 0.001 from two-tailed unpaired t-test. **f, g** BRE reporter assay in HEPG2 cells upon siLRP5/6 and siLGR4/5 knockdowns, with or without overnight BMP4 and RSPO2 treatment as indicated. n = 3 biologically independent samples. **h** BRE reporter assay in HEPG2 cells stimulated overnight by BMP4 with or without increasing amount of WNT3A treatment. WNT3A activity was validated in Supplementary Fig. 1b. n = 3 biologically independent samples. **i** Domain structures of RSPO2 and deletion mutants used in **j**. sp, signal peptide; FU, furin domain; TSP1, thrombospondin domain 1. **j** BRE reporter assay in HEPG2 cells stimulated overnight with BMP4, and with or without RSPO2 WT or FU1/2 or TSP1 deletion mutants, respectively. n = 3 biologically independent samples. Data for reporter assays (**a, b, f–h, j**) are displayed as means ± SD, and show a representative of multiple independent experiments. ns, not significant; *P < 0.05, **P < 0.01, ***P < 0.001, and ****P < 0.0001 from two-tailed unpaired t-test (**a, b, f, g, j**) or one-way ANOVA with Dunnett test (**h**). **k, l** Western blot analyses of phosphorylated Smad1 (pSmad1) and total Smad1 (tSmad1) in H1581 cells upon siRNA transfection as indicated, with 0 h, 1 h, and 2 h of BMP4 stimulation. Ratio, relative levels of pSmad1 normalized to tSmad1. Representative data from two independent experiments are shown.

glands[35–37]. In contrast, overexpression of *rspo2* failed to induce enlarged heads but instead induced *spina bifida* with reduced head structures, yielding the first indication that *rspo2* does not act by the common mode of sequestering BMP ligands (Supplementary Fig. 2c).

To confirm the *rspo2* morpholino data, we used a previously established guide RNA (gRNA)[27] to generate Crispr-Cas9-mediated *Xenopus rspo2* knockout (KO) embryos (Supplementary Fig. 3a-e). We then established gRNAs to generate Crispr-Cas9 mediated knockouts of the BMP antagonists *chordin (chd)* and

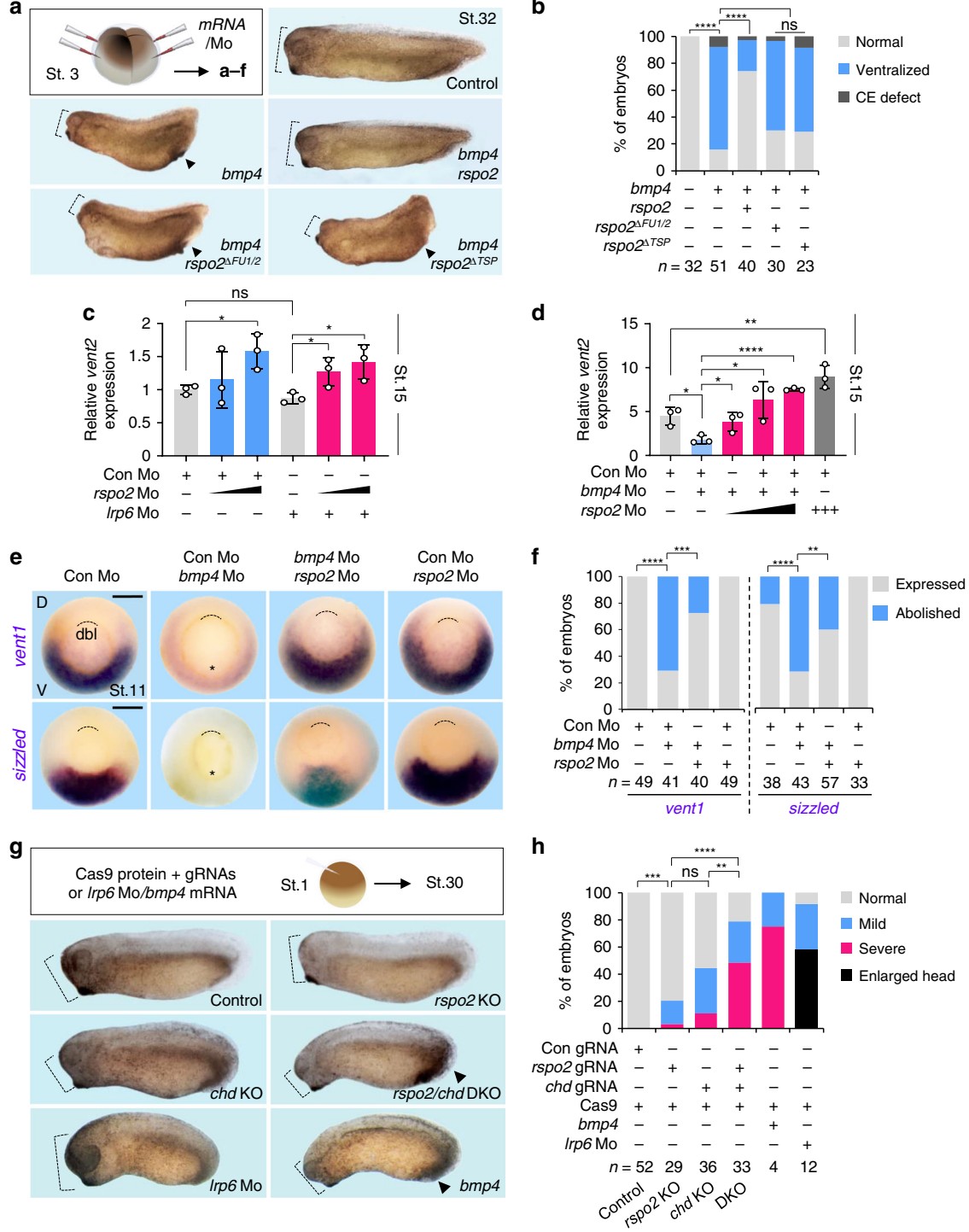

noggin (nog) (Supplementary Fig. 3a–e), whose microinjection with Cas9 protein yielded mildly ventralized embryos, which were rescued by chordin or noggin DNA, validating the specificity of the gRNAs (Supplementary Fig. 3f–i). Injection of rspo2 gRNA with Cas9 protein resulted in mildly ventralized embryos (Fig. 2g, h, Supplementary Fig. 4a, b) and increased BMP target gene (sizzled, vent1) expression, similar to knockouts of chordin or noggin (Supplementary Fig. 4c, f). Importantly, combined injection of rspo2 gRNA with either chordin or noggin gRNAs yielded strongly ventralized embryos (Fig. 2g, h, Supplementary Fig. 4a, b) and hyperactivated BMP signaling (Supplementary Fig. 4c–f). Moreover, injection of rspo3 mRNA rescued bmp4-mediated increase of

sizzled expression, suggesting that overexpressed rspo3 is also able to antagonize BMP signaling in Xenopus (Supplementary Fig. 4g, h), as in HEPG2 cells (Fig. 1a). We conclude that rspo2 is required to antagonize BMP signaling and acts in concert with BMP antagonists for proper axial patterning during Xenopus embryogenesis.

**Rspo2 is a negative feedback regulator in the Xenopus BMP4 synexpression group.** In early vertebrate embryos, genes belonging to certain signaling networks form characteristic synexpression groups, i.e., genetic modules composed of genes

**Fig. 2 Rspo2 inhibits BMP4 signaling in *Xenopus* dorsoventral embryonic patterning. a** Microinjection strategy for **a–f**, and representative phenotypes of *Xenopus laevis* tadpoles (St. 32) injected with the indicated mRNAs radially at 4-cell stage. Dashed lines, head size. Arrowheads, enlarged ventral structure. **b** Quantification of embryonic phenotypes shown in **a**. 'Ventralized' represents embryos with both small head and enlarged ventral structure, reminiscent of BMP hyperactivation. 'CE defect' refers to embryos with convergent extension (gastrulation) defects, unrelated to BMP signaling. Note that *rspo2* mRNA dosage used in **a** was below those that cause gastrulation defects. $n$ = number of embryos. **c, d** BMP-(*vent2*) reporter assays with *Xenopus laevis* neurulae (St.15) injected with reporter plasmids and the indicated Mo at 4-cell stage. $n$ = biologically independent samples and data are displayed as means ± SD. ns, not significant. *$P < 0.1$, **$P < 0.01$, ****$P < 0.0001$ from two-tailed unpaired $t$-test. **e** In situ hybridization of *vent1* and *sizzled* in *Xenopus laevis* gastrulae (St.11, dorsal to the top, vegetal view) injected as indicated. D, dorsal, V, ventral. Asterisk, abolishment of the expression. Dashed line, dorsal blastopore lip (dbl). Scale bar, 0.5 mm. **f** Quantification of embryonic phenotypes shown in (**e**). 'Expressed', normal, increased or reappearance of *vent1/sizzled* expression. 'Abolished', complete absence of *vent1/sizzled* expression. Data are pooled from two independent experiments. $n$ = number of embryos. **g** Microinjection strategy and representative phenotypes of *Xenopus tropicalis* tadpole (St.30) Crispants and tadpoles (St.30) injected with *bmp4* mRNA or *lrp6* Mo. At 1-cell stage, Cas9 protein with guide RNA (gRNA) targeting *rspo2* or *chd*, or both gRNAs were injected animally. Dashed lines, head size. Arrowheads, enlarged ventral structure. **h** Quantification embryonic phenotypes shown in **g**. 'Severe' showed small head, enlarged ventral tissues and short body axis. 'Mild' showed one or two of the defects described above. 'Normal' showed no visible differences to the uninjected control. $n$ = number of embryos. ns, not significant. **$P < 0.01$, ***$P < 0.001$, ****$P < 0.0001$ from two-tailed $\chi$2 test comparing normal versus ventralized phenotypes (**b**), two-tailed $\chi$2 test comparing expressed versus abolished (**f**), or two-tailed $\chi$2 test comparing normal versus severe and mild defects **h**.

that show tight spatio-temporal RNA coexpression and that function in the respective signaling pathway[40]. A well-characterized example is the BMP4 synexpression group, members of which are expressed like this growth factor—dorsally in the eye, heart and proctodeum of tailbud stage *Xenopus* embryos (Fig. 3a). This group consists of at least eight members, which all encode positive or negative feedback components of the BMP signaling cascade as studied in early development, including ligands, receptors and downstream components of the pathway[41]. Interestingly, we found that *rspo2* is part of the BMP4 synexpression group, being coexpressed with *bmp4* from gastrula to tadpole stages (Fig. 3a), suggesting that its expression depends on BMP signaling as for other synexpressed genes. To test this idea, we employed *Xenopus* animal cap explants, which express low levels of *rspo2* and *bmp4* to monitor *rspo2* induction upon *bmp4* overexpression (Fig. 3b). Indeed, *bmp4* induced *rspo2* expression by qRT-PCR (Fig. 3c) and in situ hybridization (Fig. 3d, e), similar to *bmp4* direct targets *sizzled* (Fig. 3c–e) and *vent1* (Fig. 3c). To test whether *rspo2* is an immediate early target of BMP4, we blocked protein synthesis with cycloheximide (CHX)[41]. Interestingly, while induction of the direct BMP4 targets *sizzled* and *vent1* by *bmp4* was unaffected by CHX, *rspo2* induction was inhibited (Fig. 3b–e). We conclude that *rspo2* is a negative feedback inhibitor within the BMP4 synexpression group and that it is an indirect BMP target gene, whose expression may depend on transcription factors of the e.g., Vent or Msx families[41,42] (Fig. 3f).

**RSPO2 and RSPO3 bind BMPR1A via the TSP1 domain to antagonize BMP signaling.** Given that RSPOs act by promoting receptor endocytosis[14,17], we postulated that RSPO2 might regulate BMP signaling through its receptors: ACVR1, BMPR1A and BMPR1B. To test this hypothesis, we analyzed the effect of RSPO1-4 treatment on BMP signaling induced by constitutively active ACVR1/BMPR1A/BMPR1B (ACVR1/BMPR1A/BMPR1B^{QD}). Interestingly, RSPO2 and RSPO3 treatment specifically inhibited BMPR1A^{QD} but not ACVR1^{QD} or BMPR1B^{QD}, while RSPO1 and RSPO4 had no effect to any of the constitutively active receptors (Fig. 4a–c).

Indeed, cell surface binding assay and in vitro binding assay revealed that RSPO2 and RSPO3, but not RSPO1 and RSPO4, bound the extracellular domain (ECD) of BMPR1A (Fig. 4d, e, Supplementary Fig. 5a). RSPO2 showed high affinity with BMPR1A ECD ($K_d \approx 4.8$ nM) (Fig. 4f), comparable to the RSPO-LGR interaction[24]. To further delineate the domains required for BMPR1A binding, we analyzed deletion mutants of

RSPO2 in cell surface binding assays with BMPR1A ECD, and found BMPR1A binding required the TSP1 but not the FU domains of RSPO2, while, conversely, LGR binding required the FU domains but not TSP1 (Supplementary Fig. 5b, c). The importance of the TSP1 domain was confirmed by in vitro binding assay showing that the isolated TSP1 domain of RSPO2, but not RSPO1, was sufficient to interact directly with BMPR1A ECD (Fig. 4g, h). Similarly, BMPR1A binding required the TSP1 domain also in RSPO3, suggesting that an analogous mode of binding applies to RSPO2 and RSPO3 (Supplementary Fig. 5d, e). Our results indicate that the specificity for the RSPO-BMPR1A interaction resides in the TSP1 domain of RSPOs. Consistently, the RSPO1 TSP1 domain shows only 43 and 50% sequence similarity to RSPO2 and RSPO3, respectively[28]. We next asked whether TSP1-domain swapping could convey BMP signaling inhibition to RSPO1. To this end, we generated a RSPO1 chimera (R1-TSP^{R2}) possessing the TSP1 domain of RSPO2 (Fig. 4i). R1-TSP^{R2} activated WNT signaling (Fig. 4j) and interacted with LGR4 (Supplementary Fig. 5f). However, unlike wild-type RSPO1, R1-TSP^{R2} bound to BMPR1A (Supplementary Fig. 5f) and antagonized BMP signaling, mimicking the effects of RSPO2 (Fig. 4k).

The importance of the TSP1 domain in BMP inhibition was further corroborated in *Xenopus*, where we took advantage of the fact that the TSP1-domain is encoded by a distinct exon in the 3′-end of the *rspo2* gene. We generated a *rspo2* Mo (*rspo2*^{ΔTSP} Mo), which specifically abolished TSP1-domain splicing, yielding 3′ truncated *rspo2* mRNA lacking the TSP1 domain but retaining the FU domains (Fig. 5a). Microinjection of *rspo2*^{ΔTSP} Mo resulted in ventralized tadpoles with shorter axis and reduced heads compared to control tadpoles, which was partially rescued by introducing a non-targeted *rspo2* mRNA (Supplementary Fig. 6a, b). *rspo2*^{ΔTSP} Morphants had no effect on WNT signaling (Fig. 5b), confirming that it does not interfere with Rspo2 FU domains that are essential for WNT activation. However, *rspo2*^{ΔTSP} Mo increased BMP signaling (Fig. 5c). Similar to *chordin* and *rspo2* Morphants, *rspo2*^{ΔTSP} Morphants showed expanded expression of the BMP target genes *vent1* and *sizzled* in gastrulae (Fig. 5d, e, Supplementary Fig. 6c, d)[38], and corresponding tadpoles were ventralized, displaying decreased *bf1* and *myoD* and increased *sizzled* expression (Fig. 5f, g)[38]. Coexpression of dominant negative *bmpr1a* (*bmpr1a*^{DN}) rescued these defects (Fig. 5d, g, Supplementary Fig. 6c, d). Taken together, these results emphasize that the TSP1 domain is a key element in providing target specificity to RSPOs, both in vitro and in vivo, and that it dictates their BMP-inhibitory function.

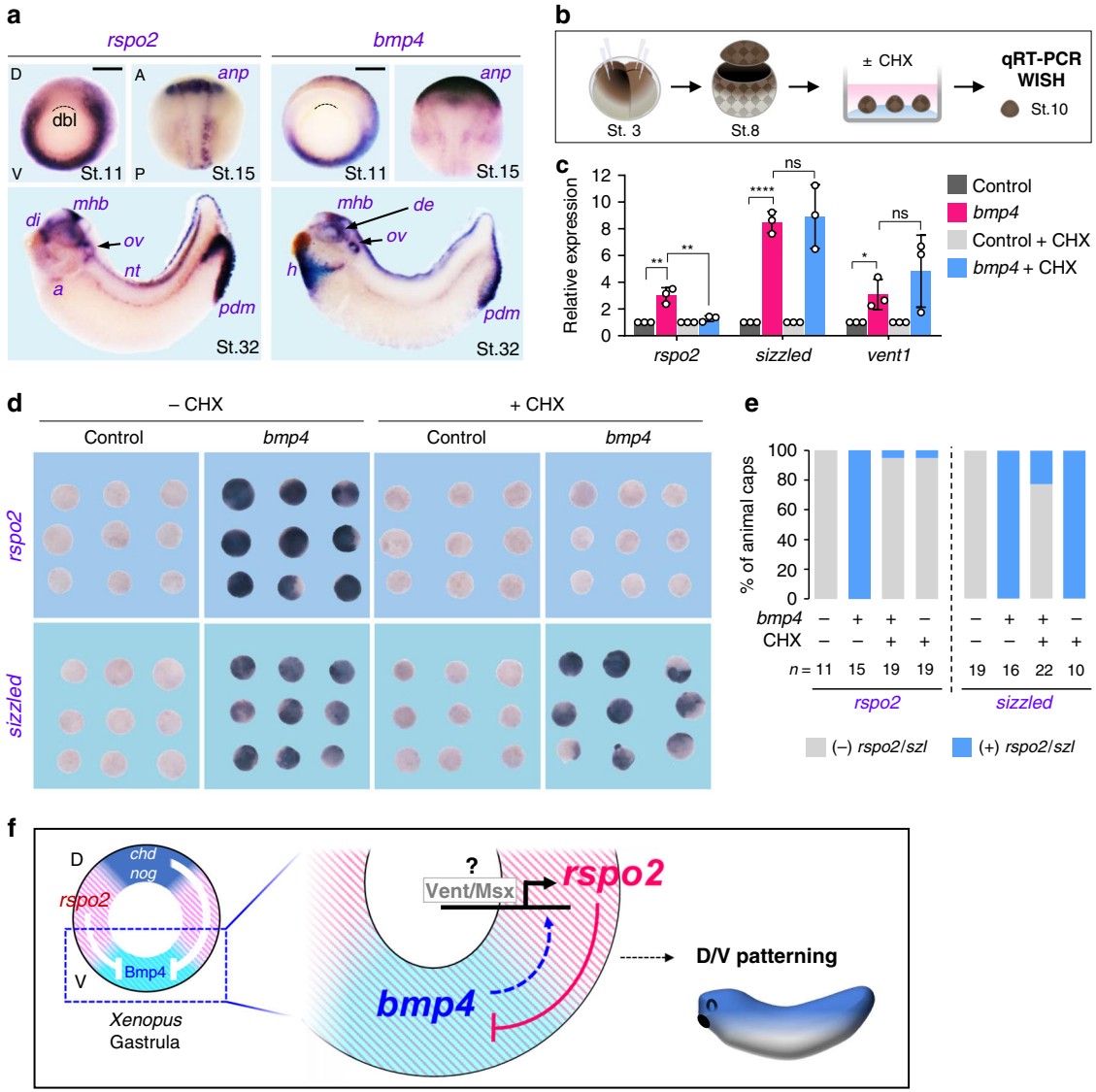

**Fig. 3 Rspo2 is a negative feedback inhibitor in the BMP4 synexpression group. a** In situ hybridization of *rspo2* and *bmp4* in *Xenopus laevis* at gastrula (St. 11, dorsal to the top, vegetal view), neurula (St. 15, anterior to the top, dorsal view), and tadpole (St. 32, anterior to the left, lateral view). Dashed lines, dorsal blastopore lip (dbl); anp, anterior neural plate; di, diencephalon; mhb, mid-hindbrain boundary; ov, otic vesicle; a, atria; nt, neural tube; de, dorsal eye; h, heart; pdm, proctodeum. Scale bar, 0.5 mm. **b** Microinjection and experimental scheme for **c**–**e**. 2 or 4 cell stage *Xenopus laevis* embryos were animally injected with control (*ppl*) or *bmp4* mRNA. The animal cap (AC) explants were dissected from injected embryos at stage 8, and either treated or untreated with cycloheximide (CHX) until control embryos reached stage 10 for qRT-PCR (**c**) or in situ hybridization (**d**, **e**). **c** qRT-PCR of *rspo2*, *sizzled*, and *vent1* expression in the AC explants injected and treated as indicated. Data are pooled from three independent experiments with similar results and displayed as means ± SD. ns, not significant, $*P < 0.05$, $**P < 0.01$, $****P < 0.0001$ from two-tailed unpaired *t*-test. **d** In situ hybridization of *rspo2* and *sizzled* in the AC explants injected and treated as indicated. **e** Quantification of (**d**). $n$ = number of the AC explants. **f** Model for Rspo2 function as a negative feedback inhibitor of BMP4 in *Xenopus* dorsoventral patterning.

**RSPO2 destabilizes the BMP receptor BMPR1A.** To investigate the consequence of RSPO-BMPR1A binding, we monitored BMPR1A protein levels upon *RSPO2* knockdown in H1581 cells and found that si*RSPO2* treatment increased BMPR1A protein levels (Fig. 6a). Similarly in *Xenopus* whole embryos, microinjection of mRNA encoding *rspo2* but not *rspo2*^ΔFU1/2 or *rspo2*^ΔTSP decreased protein levels from coinjected *bmpr1a*-EYFP mRNA (Fig. 6b). Immunofluorescence microscopy (IF) of *Xenopus* animal cap explants showed that Bmpr1a-EYFP localizes to the plasma membrane, where it was once again reduced by *rspo2* but not by *rspo2*^ΔFU1/2 or *rspo2*^ΔTSP mRNA (Fig. 6c–e). Focusing on *Xenopus* ventrolateral marginal zone (VLMZ) explants, where endogenous *rspo2*, *bmpr1a* and *bmp4* are

coexpressed, showed that ablation of *rspo2* by Mo injection results in significant increase of Bmpr1a-EYFP plasma membrane levels (Fig. 6f–h). Moreover, in VLMZ from *rspo2*^ΔTSP Morphants, Bmpr1a levels were also increased (Fig. 6f–h), which was confirmed by western blot analysis (Fig. 6i). Altogether, our results suggest that RSPO2 destabilizes BMPR1A.

**RSPO2 requires ZNRF3 to antagonize BMP receptor signaling.** We next turned to the role of the FU domains in RSPO2, which are also required for inhibition of BMP signaling (Figs. 1j, 2a, b and 6b–e). FU1 and FU2 domains confer RSPO binding to ZNRF3/RNF43 and LGRs, respectively[27]. Since our results demonstrated an LGR-independent mode of action (Fig. 1f), and

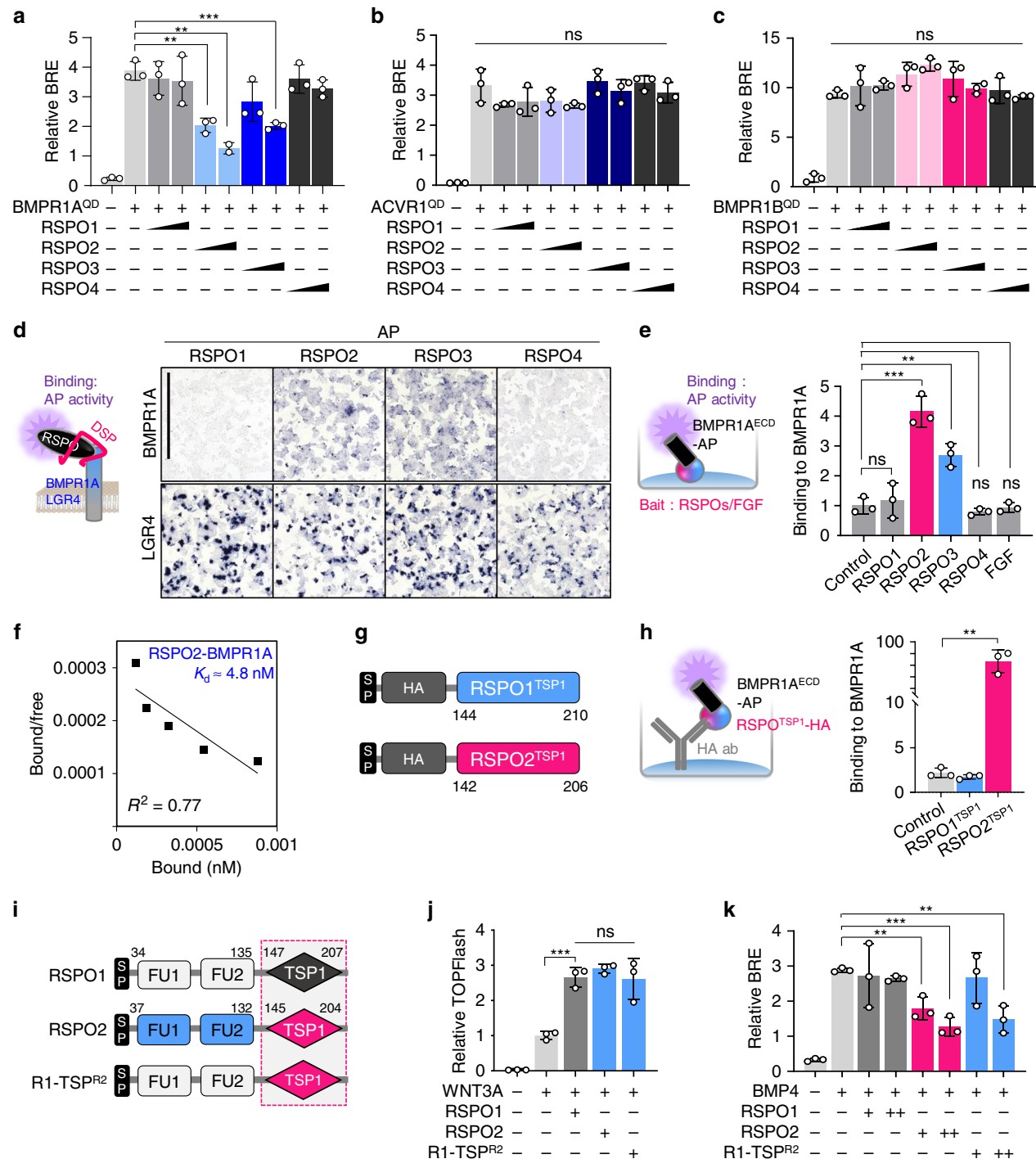

since *rspo2* destabilized Bmpr1a (Fig. 6b), we hypothesized that RSPO2 acts via ZNRF3/RNF43 E3 ligases to interfere with BMPR1A. ZNRF3 and RNF43 were both expressed in HEPG2 and H1581 cells, and could be significantly knocked down by siRNA (Supplementary Fig. 7a). Knockdown of *ZNRF3/RNF43* (Fig. 7a) or expression of a dominant negative ZNRF3 (ZNRF3$^{\Delta R}$)[26] (Fig. 7b) prevented inhibition of BMP signaling by RSPO2 in HEPG2 cells, supporting that RSPO2 requires ZNRF3/ RNF43 to antagonize BMP signaling. In *Xenopus*, *znrf3* was broadly expressed from gastrula stages onwards, like *bmpr1a* (Supplementary Fig. 7b, c). *znrf3* ablation by Mo elicited head and axis defects that were rescued by coinjection of human *ZNRF3* mRNA, as previously described[43] (Supplementary Fig. 7d, e).

Interestingly, *znrf3* Morphants at neurula showed increased BMP signaling by BMP-reporter assay and *rspo2* mRNA coinjection could not reduce it (Fig. 7c). Moreover, IF in *Xenopus* animal cap explants showed that *rspo2*-induced destabilization of Bmpr1a protein levels was prevented by *ZNRF3$^{\Delta R}$* (Fig. 7d, e). Altogether, these results support that to function as BMP antagonist, RSPO2 requires ZNRF3.

**RSPO2 requires the FU1 but not FU2 domain to antagonize BMP signaling.** To corroborate that to function as BMP antagonist, RSPO2 depends on ZNRF3/RNF43, but not on LGRs, we next generated deletion mutants of the FU1 and FU2 domains in human RSPO2, which mediate binding to ZNRF3/RNF43 and

**Fig. 4 RSPO2 and RSPO3 interact with BMPR1A via the TSP1 domain. a–c** BRE reporter assays in HEPG2 cells transfected with constitutively active (QD) BMPR1A (**a**), ACVR1 (**b**), or BMPR1B (**c**) with or without BMP4 and RSPO1-4 treatment overnight. **d** Cell surface binding assay in HEK293T cells. (Left) Scheme of the assay. Cells were transfected with BMPR1A or LGR4 DNA, and treated with same amount of RSPO1-4-AP upon DSP crosslinking as indicated. Binding was detected as purple stain on cell surface by chromogenic AP assay. (Right) Images of cells transfected and treated as indicated. Data shows a representative from four independent experiments. For quantification, see Supplementary Fig. 5a. Scale bar, 1 mm. **e** In vitro binding assay between RSPO1-4, FGF and BMPR1A$^{ECD}$. (Left) Scheme of the assay. RSPOs and FGF recombinant proteins were coated on plate as baits, followed by BMPR1A$^{ECD}$-AP treatment overnight. (Right) Bound BMPR1A$^{ECD}$ was detected by chromogenic AP assay. Normalized AP activity with control treatment was set to 1. **f** Scatchard plot of RSPO2 and BMPR1A$^{ECD}$ binding to validate $K_d$ for RSPO2-BMPR1A. **g** Domain structures of the RSPO1 and RSPO2$^{TSP1}$ with Strep-HA and flag tags used in **h**. SP, signal peptide; TSP1, thrombospondin domain 1. **h** In vitro binding assay for RSPO$^{TSP1}$ and BMPR1A$^{ECD}$. (Left) Scheme of the assay. HA-harboring RSPO1/2$^{TSP1}$ were captured to HA antibody coated plate, and BMPR1A$^{ECD}$-AP was treated overnight. (Right) Bound BMPR1A to RSPO$^{TSP1}$ was detected with absorbance. **i** Domain structures of the RSPO1, RSPO2, and R1-TSP$^{R2}$. SP, signal peptide; FU, furin domain; TSP1, thrombospondin domain 1. Dashed box indicates the TSP1 domain swapping. **j** TOPFlash reporter assay in HEPG2 cells upon WNT3A with or without (**i**) as indicated. **k** BRE reporter assay in HEPG2 cells upon BMP4 with or without (**i**) as indicated. For reporter assays (**a–c**, **j**, **k**), *n* = 3 biologically independent samples; In vitro binding assays (**e**, **h**), *n* = 3 experimentally independent samples. All data are displayed as mean ± SD. ns, not significant, *$P < 0.05$, **$P < 0.01$, ***$P < 0.001$, ****$P < 0.0001$ from two-tailed unpaired *t*-test (**a**, **b**, **e**, **h**, **j**, **k**) or one-way ANOVA test (**c**).

LGRs, respectively[27] (Supplementary Fig. 8a). RSPO2$^{ΔFU1}$ lost ZNRF3 binding (Supplementary Fig. 8b), yet it bound LGR4 (Supplementary Fig. 8c), but did not inhibit BMP4 signaling (Fig. 7f). Conversely, RSPO2$^{ΔFU2}$ bound ZNRF3 but not to LGR4 (Supplementary Fig. 8b, c), yet it still antagonized BMP4 signaling (Fig. 7g). To corroborate LGR-independent function in vivo, we generated *Xenopus* Rspo2$^{ΔFU1}$ and FU2 point mutant Rspo2$^{F107E}$ (Supplementary Fig. 8d)[17], which displayed ZNRF3 and LGR4 binding characteristics like human RSPO2 mutants (Supplementary Fig. 8e, f). IF in *Xenopus* animal cap explants injected with *bmpr1a*-EYFP and either *rspo2* wildtype or *rspo2* mutants confirmed that FU1 but not FU2 deletion eliminates the ability of Rspo2 to remove plasma membrane Bmpr1a (Fig. 7h, i). Taken together, our results clearly indicate that the FU1 mediated ZNRF3/RNF43 binding is crucial while FU2 mediated LGR binding is dispensable for RSPO2 to antagonize BMP receptor signaling.

**RSPO2 bridges BMPR1A and ZNRF3 and triggers BMP receptor clearance from the cell surface.** The interaction of RSPO2 and RSPO3 with BMPR1A, as well as ZNRF3, suggested that R-spondins bridge both transmembrane proteins. In vitro binding assays (Fig. 8a, b) and colocalization by IF (Fig. 8c, d, Supplementary Fig. 9a, b), confirmed that ZNRF3 interacted with BMPR1A in the presence of RSPO2 or RSPO3 but not of RSPO1. Emphasizing once again the importance of the FU1 and TSP1 domains for this interaction, in vitro ZNRF3-BMPR1A-RSPO2 ternary complex formation was prevented by TSP1, FU1/2, or FU1 deletion (Supplementary Fig. 9c–g), whereas it remained intact upon FU2 deletion (Supplementary Fig. 9h).

Since ZNRF3/RNF43 eliminate WNT receptors from the cell surface by co-internalization and lysosomal degradation[25,26], we considered an analogous function in BMPR1A turnover. We monitored BMPR1A localization by IF in H1581 cells and found that it was absent from the plasma membrane but abundantly colocalized with ZNRF3 in cytoplasmic vesicles (Fig. 8e, i), suggesting that it may be internalized by endogenous RSPO2. Indeed, upon knockdown of *RSPO2*, but not *LRP6* or *LGR4/5*, BMPR1A accumulated at the plasma membrane (Fig. 8f–i). Importantly, IF (Fig. 8j–m) and cell surface biotinylation assays (Fig. 8n) showed that upon *ZNRF3/RNF43* siRNA treatment, BMPR1A also accumulated at the plasma membrane.

To test if RSPO2/ZNRF3 target BMPR1A for endocytosis and lysosomal degradation, we treated cells with the clathrin inhibitor monodansylcadaverin (MDC), which eliminated inhibition of BMP signaling by RSPO2 (Fig. 8o). In addition, si*RSPO2* abolished the colocalization of BMPR1A with the early endosome marker EEA1 (Fig. 8p–q) and lysosomal marker Lamp1

(Fig. 8r–s), suggesting that RSPO2 binding promotes BMPR1A internalization and degradation via ZNRF3 ternary complex formation. Consistently, 20 min exposure to RSPO2 increased internalized BMPR1A in cell surface biotinylation assays in H1581 cells (Supplementary Fig. 10a) and induced vesicular Bmpr1a-EYFP in *Xenopus* animal caps (Supplementary Fig. 10b, c). Taken together, our results support a model (Supplementary Fig. 10d) wherein RSPO2 bridges ZNRF3 and BMPR1A and routes the ternary complex towards clathrin-mediated endocytosis for lysosomal degradation, thereby antagonizing BMP signaling. We suggest that a similar mechanism applies to RSPO3 but not RSPO1 and RSPO4.

## Discussion

The three main findings of our study are (i) the discovery R-spondins as BMP receptor antagonists, (ii) that RSPO2 depletes BMPR1A/ALK3 by engaging ZNRF3 for internalization and lysosomal degradation, and (iii) that in *Xenopus*, *rspo2* is a negative feedback inhibitor of the BMP4 synexpression group, which cooperates with Spemann organizer effectors to inhibit BMP signaling during axis formation. Given the importance of RSPOs and BMPs as developmental regulators, as well as growth factors of normal and malignant stem cells, these conclusions have implications for development and cancer.

With regard to stem cells, R-spondins are a key ingredient of the culture media, which have made the organoid revolution possible[21,22] and their rational use requires an understanding of their mechanism of action. For example, the fact that R-spondins inhibit BMP signaling may explain the reported non-equivalence of WNT and RSPO ligands in stem cells and development[31–33]. It may also explain their potency as stem cell growth factors, as e.g., intestinal stem cells requires both, WNT activation and BMP inhibition[21,22].

TGFβ growth factors play an eminent role in biology and medicine, and their receptor signaling is exquisitely regulated extracellularly with over 20 TGFβ antagonists, most of which antagonize signaling by ligand sequestration (e.g., Cerberus, Chordin, Follistatin, Gremlin, Noggin, and Sost)[1,11]. Two extracellular BMP receptor antagonists are known, BMP3 and Inhibin[12,13]. Both are TGFβ family members, whose unproductive binding to type II receptors prevents signal transmission. Relatedly, the BMP antagonist BAMBI is a BMP pseudoreceptor lacking kinase activity, which also leads to formation of a dead-end complex with BMP receptors[44]. In contrast, RSPO2 and RSPO3 share no sequence homology with TGFβ family members, they inhibit type I instead of type II BMP receptors, and they do so by engaging the ZNRF3 E3 transmembrane ubiquitin ligase to internalize BMPR1A. RSPO2 thereby routes BMPR1A to

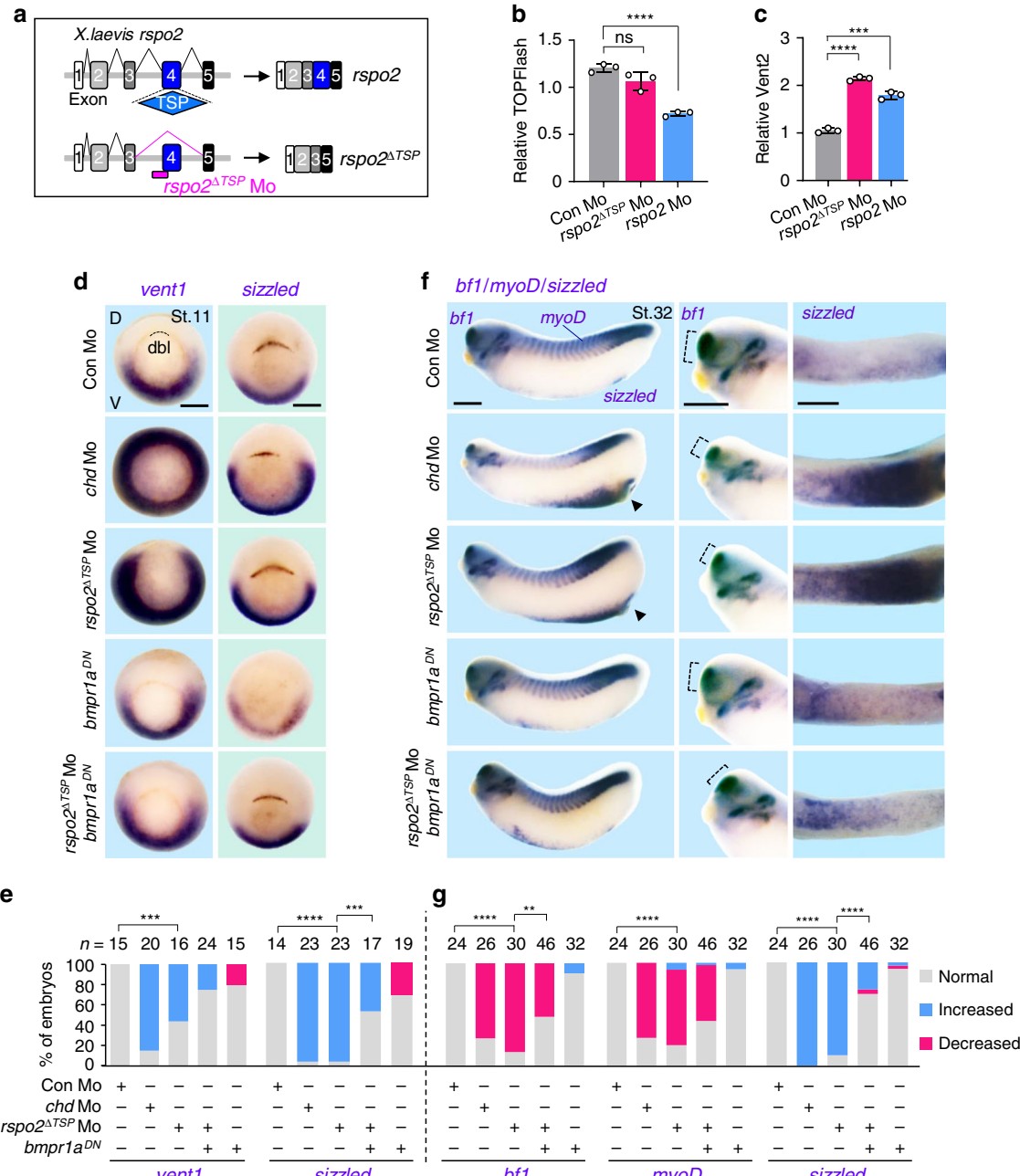

**Fig. 5 Loss of Rspo2-TSP1 domain activates BMP signaling in *Xenopus* development. a** Scheme for *rspo2*$^{\Delta TSP}$ splicing Mo in *Xenopus laevis*. **b** TOPFlash assay in *Xenopus laevis* neurulae (St.15) injected radially at 4-cell stage with reporter plasmids and Mo as indicated. Data are displayed as mean ± SD; ns, not significant, ****$P < 0.0001$ from two-tailed unpaired $t$-test. $n = 3$ biologically independent samples. **c** BMP-reporter (*vent2*) assay in *Xenopus laevis* neurulae (St.15) injected radially at 4-cell stage with reporter plasmids and Mo as indicated. Data are displayed as mean ± SD; ***$P < 0.001$, ****$P < 0.0001$ from two-tailed unpaired $t$-test. $n = 3$ biologically independent samples. **d–g** In situ hybridization of BMP4 targets *vent1* and *sizzled* in *Xenopus laevis*. Embryos were injected radially and equatorially at 4-cell stage as indicated. Gastrulae (St.11) (**d**) and quantification (**e**); Tadpoles (St. 32) (**f**) and quantification (**g**). Dashed lines, dorsal blastopore lip (dbl) (**d**) or *bf1* expression (**e**); D, dorsal; V, ventral. For **f**, left, lateral view; middle, magnified view of head; right, magnified view of ventral side. 'Increased/Decreased' represents embryos with significant expansion/reduction of *sizzled* or *vent1* signals toward the dorsal/ventral side of the embryo (**e**), or with significant increase/decrease of the signal strength (**g**). ns, not significant. $n$, number of embryos. Scale bar, 0.5 mm. Scoring of the embryos for quantification was executed with blinding from two individuals. For **e**, **g**, **$P < 0.01$, ***$P < 0.001$, ****$P < 0.0001$ from two-tailed $\chi2$ test comparing normal versus increased. Data are pooled from at least two independent experiments.

clathrin-mediated endocytosis for lysosomal degradation. This mode of action resembles the function of the Spastic Paraplegia related gene NIPA1, a transmembrane antagonist, which promotes BMP receptor type II endocytosis and lysosomal degradation[45].

Other type I BMP receptors besides BMPR1A include ACVRL1, ACVR1, and BMPR1B[7]. However, we found that RSPO2 specifically binds to BMPR1A but not to ACVR1 or BMPR1B (Supplementary Fig. 11), which explains why ACVR1 and BMPR1B signaling were not antagonized by RSPO2 in

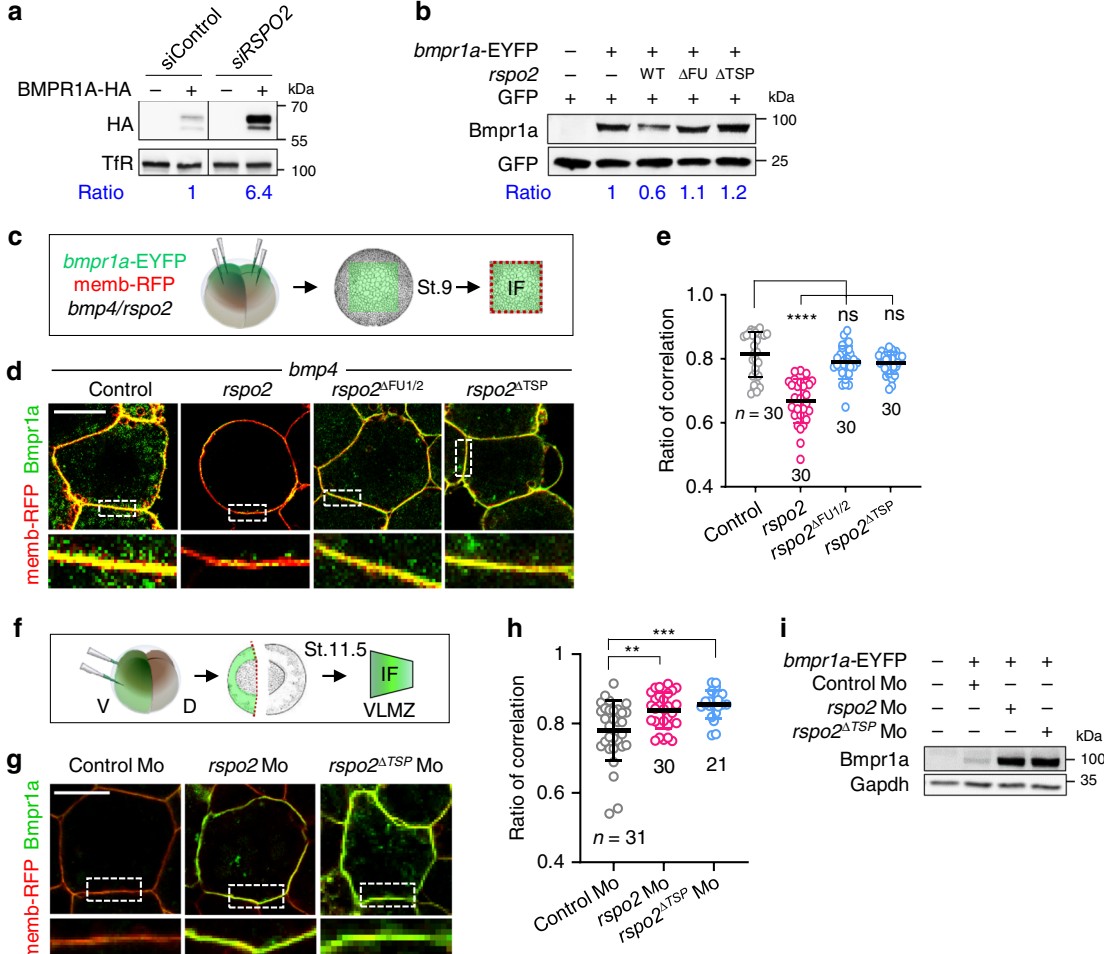

**Fig. 6 RSPO2 removes cell surface BMPR1A. a** Western blot analysis in H1581 cells treated with siControl or si*RSPO2* as indicated and transfected with or without BMPR1A-HA DNA. Transferrin receptor (TfR), a loading control. Ratio, relative levels of BMPR1A-HA normalized to TfR. Data shows representative result from 3 independent experiments with similar conclusion. **b** Western blot analysis of Bmpr1a in *Xenopus laevis* (St. 15) neurulae injected animally at 2-cell to 4-cell stages as indicated. GFP mRNA was injected as an injection control. Ratio, relative levels of Bmpr1a normalized to GFP. Data shows representative result from 3 independent experiments with similar conclusion. **c** Scheme for immunofluorescence microscopy (IF) in *Xenopus laevis* animal cap (AC) explants. Embryos were injected animally at 4-cell stage with *bmpr1a*-EYFP and memb-RFP mRNA along with *bmp4* and *rspo2* wild-type or mutant mRNA. AC explants were dissected at St.9 for IF. Membrane (memb)-RFP was used as a control comparing relative change of Bmpr1a-EYFP signal at cell surface. **d** IF for Bmpr1a (green) and cell membrane (red) in AC explants injected as indicated, with a representative cell (top) and magnification (inset). Scale bar, 20 μm. **e** Quantification of **d**. *n* = the number of areas analyzed and data are displayed as mean ± SD. ns, not significant; ****$P < 0.0001$ from two-tailed unpaired *t*-test. **f** Scheme for IF in *Xenopus laevis* ventrolateral marginal zone explants (VLMZ). Embryos were ventrally injected at 4-cell stage with *bmpr1a*-EYFP and memb-RFP mRNA with Mo. VLMZs were dissected at stage 11.5 for IF. **g** IF for Bmpr1a (green) and cell membrane (memb-RFP, red) in VLMZ injected with mRNA and Mo as indicated. Scale bar, 20 μm. **h** Quantification of **g**. *n* = the number of areas analyzed and data are displayed as mean ± SD. **$P < 0.01$, ***$P < 0.001$ from two-tailed unpaired *t*-test. **i** Western blot analysis of Bmpr1a in *Xenopus laevis* neurulae (St. 18) injected radially at 4-cell stage as indicated. Gapdh, a loading control. Data shows representative result from 2 independent experiments with similar conclusion.

human cells (Fig. 4a–c). Consistently, BMPR1A and BMPR1B only show 42% identity in their extracellular domain[46].

BMPR1A engages not only various BMPs but also GDFs[1], and hence RSPO-mediated inhibition may potentially affect signaling in multiple contexts. On the other hand, the specificity of RSPO2 for BMPR1A may provide therapeutic opportunities on the background of pleiotropic BMP ligands effects.

RSPO2 engages ZNRF3 to antagonize BMP signaling, implying that ZNRF3 is also a negative regulator not only of WNT, but also BMP signaling. Consistently, our results indicate that loss of *ZNRF3* increases BMP signaling. Moreover, *ZNRF3* over-expression induces expression of Spemann organizer genes in *Xenopus* embryos, which is characteristic not only for WNT but also BMP inhibition[43]. In WNT signaling, the role of RSPO2 is to protect WNT receptors from ubiquitination and internalization

by ZNRF3, by forming a ternary complex with LGR4-6 and triggering endocytosis. In contrast, during BMP signaling, RSPO2 directly forms a ternary complex with ZNRF3-BMPR1A to internalize and degrade the type I receptor. Our data also imply a possible function of RNF43 in antagonizing BMP signaling, inviting a closer inspection of its loss-of-function phenotypes[25,26].

A number of studies emphasized the importance of the Furin domains in RSPOs, which are necessary and sufficient for activation of WNT signaling[17,20,26,28], however, the role of the TSP1 domain has received less attention. We found that the specificity of RSPOs for BMP signaling is dictated by the TSP1 domain, which binds directly to BMPR1A. Unlike RSPO2 and RSPO3, RSPO1 and RSPO4 do not inhibit BMP signaling, the key difference residing in the TSP1 domain, as domain swapping of the TSP1 domain is sufficient to confer BMP inhibition upon RSPO1.

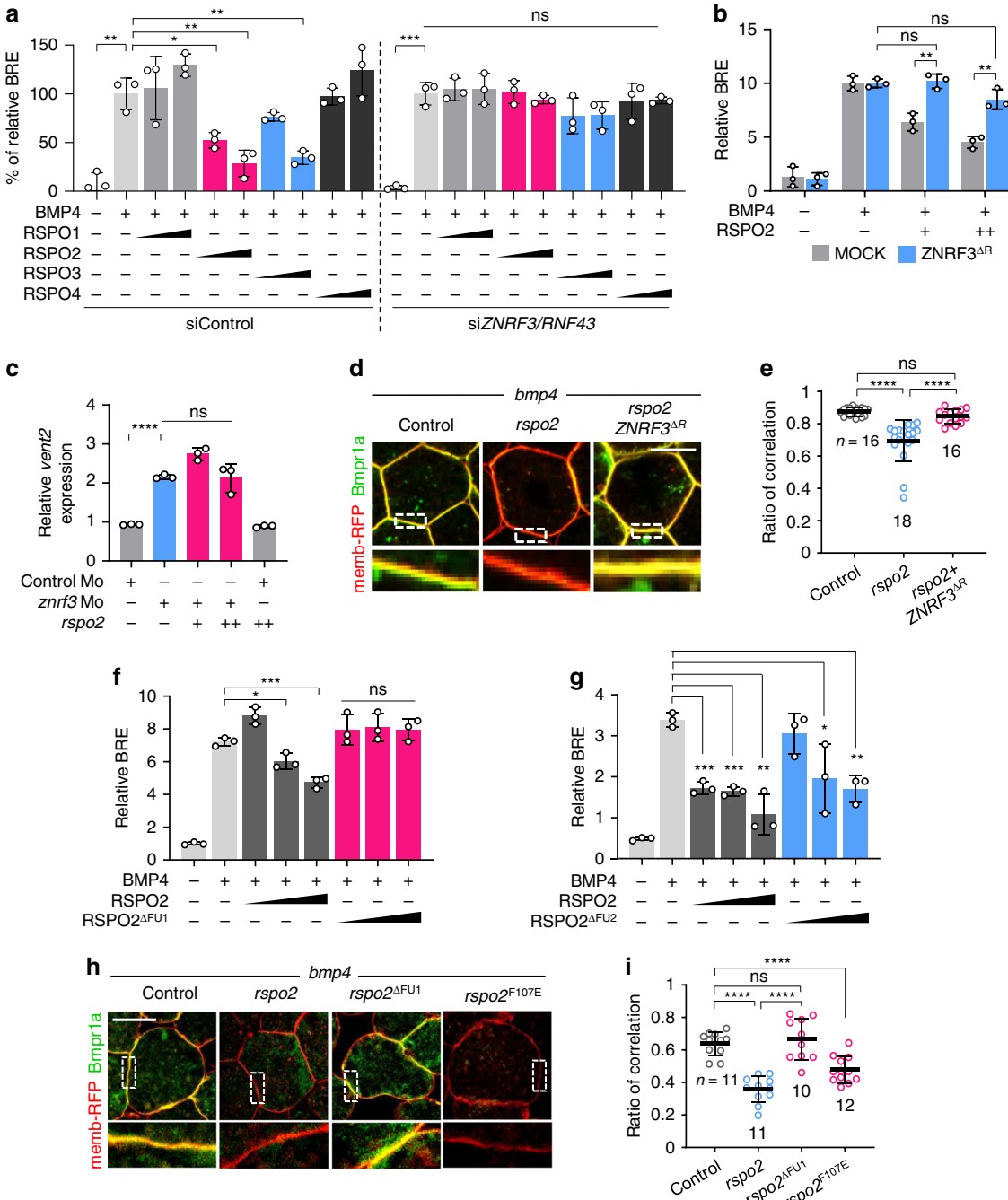

**Fig. 7 RSPO2 requires ZNRF3 to antagonize BMP4-BMPR1A signaling. a** BRE reporter assay in HEPG2 cells. Cells were transfected with siControl or si*ZNRF3*/si*RNF43*, and BMP4 with or without RSPO1-4 were added overnight as indicated. Normalized BRE activity upon BMP4 without RSPO2 stimulation was set to 100%. *n* = 3 biologically independent samples. **b** BRE reporter assay in HEPG2 cells upon ZNRF3$^{\Delta R}$ transfection, with or without overnight BMP4 and RSPO2 treatment as indicated. *n* = 3 biologically independent samples. **c** BMP-reporter (*vent2*) assay in *Xenopus laevis* St.15 neurulae. Embryos were injected animally with reporter plasmids and the indicated Mo with or without *rspo2* mRNA at 4-cell stage. Normalized *vent2* activity of control Mo injected embryos with reporter plasmids was set to 1. *n* = 3 biologically independent samples. **d** Immunofluorescence microscopy (IF) in *Xenopus laevis* animal cap explants for Bmpr1a (green) and the plasma membrane (red) from embryos injected with mRNA as indicated, with a representative cell (top) and magnification (inset). Scale bar, 20 μm. For scheme, see Fig. 6c. **e** Quantification of **d**. *n* = areas analyzed and data are displayed as mean ± SD. ns, not significant, ****$P < 0.0001$ from one-way ANOVA analysis. **f-g** BRE reporter assay in HEPG2 cells treated with BMP4 and RSPO2/RSPO2$^{\Delta FU1}$/RSPO2$^{\Delta FU2}$ overnight as indicated. For domain structure of RSPO2$^{\Delta FU1}$/RSPO2$^{\Delta FU2}$, see Supplementary Fig. 8a. *n* = 3 biologically independent samples. **h** IF for Bmpr1a (green) and plasma membrane (red) in animal cap explants injected as indicated, with a representative cell (top) and magnification (inset) showing the plasma membrane. Scale bar, 20 μm. For domain structure of *Xenopus* Rspo2 mutants, see Supplementary Fig. 8d. **i** Quantification of **h**. *n* = areas analyzed and data are displayed as mean ± SD. ns, not significant, ****$P < 0.0001$ from one-way ANOVA analysis. All data are displayed as mean ± SD. The *P* values for reporter assays (**a-c**, **f**, **g**) were determined from two-tailed unpaired *t*-test. ns, not significant; *$P < 0.05$, **$P < 0.01$, ***$P < 0.001$, ****$P < 0.0001$.

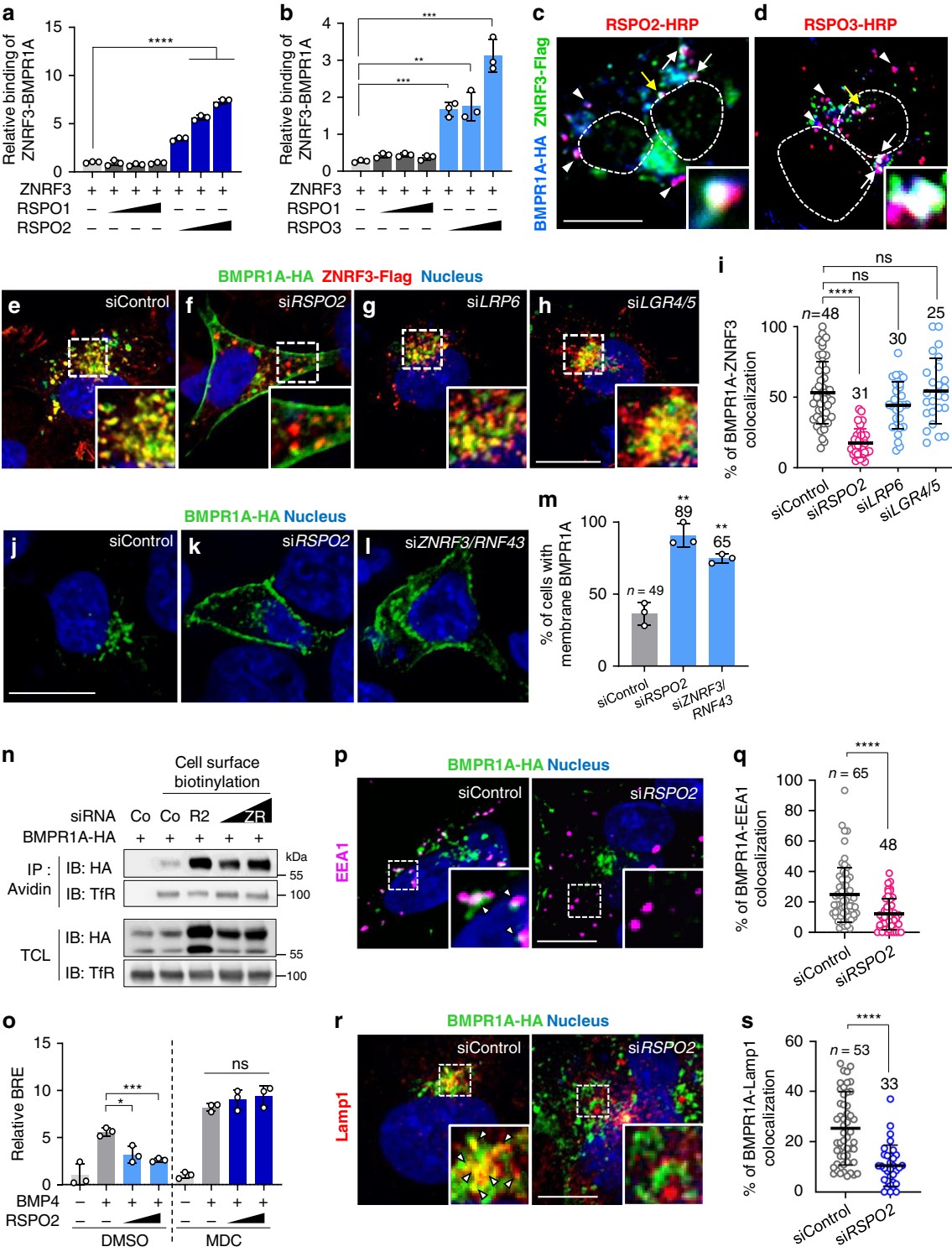

The physiological role in vivo is highlighted by *rspo2* Morphants specifically lacking the TSP1 domain, which displayed phenotypic defects due to BMP hyperactivation (Fig. 5). The TSP1 domain also binds to heparin sulfate proteoglycans (HSPG) e.g., syndecans (SDC)[27,29,30], which raises the possibility of cooperation between RSPOs and SDC in BMP receptor regulation. Indeed, SDC1 and SDC3 have been implicated as negative regulators in BMP signaling, but the underlying mechanisms remained unclear[47,48]. Hence, it will be interesting to investigate the role of SDCs in BMPR1A-RSPO interactions. HSPGs are also

coreceptors in FGF signaling, which may explain why mis-expressed *rspo2* can inhibit FGF signaling in *Xenopus* animal cap explants[49].

We established that *Xenopus* Rspo2 cooperates with Noggin and Chordin released by the Spemann organizer in repressing BMP signaling to modulate the BMP morphogen gradient, which controls axial patterning. Yet, overexpression of *rspo2* unlike of *noggin* and *chordin*, does not strongly dorsalize early embryos. The reason is that instead of sequestering BMP ligands, RSPO2 specifically targets the BMP receptor BMPR1A in early

**Fig. 8 RSPO2 bridges BMPR1A and ZNRF3 and triggers BMP receptor clearance from the cell surface. a, b** In vitro binding assay between ZNRF3 and BMPR1A$^{ECD}$ mediated by RSPO1-3. ZNRF3-Fc protein was used as a bait, with sequential RSPO1-3 protein and BMPR1A$^{ECD}$-AP treatment. Bound BMPR1A$^{ECD}$ to ZNRF3 was detected by chromogenic AP assay. $n = 3$ experimentally independent samples. **c, d** IF in H1581 cells transfected with BMPR1A-HA and ZNRF3-flag DNA upon RSPO2 and RSPO3-HRP treatment for 3 h. RSPOs (red) were visualized with tyramid signal amplification. BMPR1A (blue) and ZNRF3 (green) were stained against HA and flag antibody. White arrowheads, colocalized BMPR1A/RSPO2; white arrows, colocalized BMPR1A/RSPO2-3/ZNRF3; yellow arrow, colocalized BMPR1A/RSPO2-3/ZNRF3 in magnified inset; Dashed lines, nucleus. Scale bar, 20 μm. **e–h** IF of colocalized BMPR1A (green)/ZNRF3 (red) in H1581 cells treated with siRNA as indicated. Nuclei were stained with Hoechst. Scale bar, 20 μm. **i** Quantification of BMPR1A colocalizing with ZNRF3 from **e–h**. **j–l** IF of BMPR1A (green) in H1581 cells treated with siRNA as indicated. **m** Quantification of cells harboring membrane localized BMPR1A from **j–l**. Scale bar, 20 μm. **n** Cell surface biotinylation assay in H1581 cells treated with BMPR1A-HA and siRNA as indicated. Co, control; R2, RSPO2; ZR, ZNRF3/RNF43 siRNA. After labeling surface proteins with Biotin, lysates were pulled down with streptavidin beads and subjected to Western blot analysis. Transferrin receptor (TfR), a loading control. TCL, Total cell lysate. Data shows results representative for 2 from 3 independent experiments. **o** BRE reporter assay in HEPG2 cells treated as indicated. MDC, monodansylcadaverin. $n = 3$ biological replicates. **p** IF of colocalized BMPR1A (green)/EEA1 (magenta) in H1581 cells treated with siRNA. White arrowheads, colocalized BMPR1A/EEA1 in magnified inset. **q** Quantification of **p**. Scale bar, 20 μm. **r** IF of colocalized BMPR1A (green)/Lamp1 (red) in H1581 cells treated with siRNA as indicated. White arrowheads, colocalized BMPR1A/Lamp1 in magnified inset. Scale bar, 20 μm. **s** Quantification of **r**. For **i, m, q, s**, $n =$ the number of cells pooled from 2 independent experiments. Data for (**a, b, i, m, o, q, s**) are displayed as mean ± SD. ns, not significant, $*P < 0.05$, $**P < 0.01$, $***P < 0.001$, $****P < 0.0001$ from two-tailed unpaired $t$-test.

*Xenopus* embryos and that BMPR1A and BMPR1B play overlapping roles in dorsoventral patterning[50,51]. Also unlike *noggin* and *chordin*, *rspo2* is not expressed in the organizer but is a negative feedback inhibitor of the BMP4 synexpression group, similar to the BMP pseudoreceptor *bambi*[44,52]. Like *bambi*, *rspo2* is an indirect BMP4 target gene, which may require Vent or Msx transcription factors for expression. Negative feedback in BMP signaling expands the dynamic BMP signaling range essential for proper embryonic patterning and reduce inter-individual phenotypic and molecular variability in *Xenopus* embryos[42]. Indeed, *rspo2* deficiency by itself has only mild effects on axis formation and dorsoventral marker gene expression, while defects manifest upon misbalance of BMP signaling (*bmp4*-overexpression, *noggin/chordin* knockdown). Functional redundancy between BMP antagonists is a characteristic feature observed in fish, frog, and mouse embryos[53–56]. We note that the mouse *Rspo2* expression pattern at E9.5 mimics that of mouse *Bmp4*, including forebrain, midbrain/hindbrain junction, branchial arches and limb apical ectodermal ridge[57,58]. Thus, although *Rspo2* deficient mouse embryos gastrulate normally[59], it may be fruitful to analyze compound mutants between *Rspo2* and BMP antagonists for axial defects. The fact that R-spondins are bifunctional ligands, which activate WNT signaling and inhibit BMP signaling has implications for development, stem cell biology, and cancer. Mechanistically, the general picture emerging is that R-spondins function as adapters, which escort client extracellular proteins for ZNRF3/RNF43-mediated degradation, e.g., LGR4-6 and BMPR1A. Our results assign a key role to the largely ignored TSP1 domain of R-spondins in providing target specificity. The substantial sequence variability between TSP1 domains of RSPO1-4 invites screening for additional RSPO receptor targets beyond BMPR1A.

## Methods

**Cell lines and growth conditions.** HEK293T and HEPG2 cells (ATCC) were maintained in DMEM High glucose (Gibco 11960) supplemented with 10% FBS (Capricorn FBS-12A), 1% penicillin-streptomycin (Sigma P0781), and 2 mM L-glutamine (Sigma G7513). H1581 cells (gift from Dr. R. Thomas) were maintained in RPMI (Gibco 21875) with 10% FBS, 1% penicillin-streptomycin, 2 mM L-glutamine and 1 mM sodium pyruvate (Sigma S8636). Mycoplasma contamination was negative in all cell lines used.

***Xenopus laevis* and *Xenopus tropicalis*.** *Xenopus laevis* frogs were obtained from Nasco. *Xenopus tropicalis* frogs were obtained from Nasco, National *Xenopus* Resource (NXR) and European *Xenopus* Resource Centre (EXRC).

All *X.laevis* and *X.tropicalis* experiments were approved by the state review board of Baden-Württemberg, Germany (permit number 35-9185.81/G-141/18 (Regierungspräsidium Karlsruhe)) and executed according to federal and institutional guidelines and regulations. Developmental stages of the embryos were determined according to Nieuwkoop and Faber (Xenbase). No statistical analysis was done to adjust sample size before the experiments. No randomization of injection order was used during the experiments.

**Constructs.** Alkaline phosphatase (AP) fusions with RSPOs (human RSPO1$^{\Delta C}$-AP-pCDNA3, RSPO2$^{\Delta C}$-AP-pCDNA3, RSPO2$^{\Delta C}$-AP-pCS2+, RSPO3$^{\Delta C}$-AP-pCDNA3, murine RSPO4$^{\Delta C}$-pCDNA3) were generated by replacing the C-terminal domain ($\Delta C$) by AP and used to produce conditioned media. Human RSPO2 wild-type (RSPO2), the Furin1 and the Furin2 domain deletion mutants (RSPO2$^{\Delta FU1/2}$), and the TSP1 domain deletion mutant (RSPO2$^{\Delta TSP}$) are ORFs lacking the C-terminal domain, C-terminally tagged with a Flag-tag and subcloned into pCS2+[20]. R1-TSP$^{R2}$, R1-TSP$^{R2}$-AP and R1-TSP$^{R2}$-Flag plasmids were cloned in pCS2+. Human RSPO2$^{\Delta FU1}$ (deletion of amino acids encompassing the 6 cystines in the FU1 domain) and RSPO2$^{\Delta FU2}$ mutants (deletion of amino acids encompassing the 8 cystines in the FU2 domain) were cloned in Flag-tag or AP-tag pCS2+. Human RSPO1$^{TSP1}$ and RSPO2$^{TSP1}$-HA were cloned in Streptag-HA-flag-pCS2+. *Xenopus* Rspo2$^{\Delta FU1}$ (deletion of amino acids encompassing the 6 cystines in the FU1 domain) and Rspo2$^{F107E}$ mutants were cloned in Myc-tag or AP-tag pCS2+. All constructs were confirmed by sequencing. Conditioned media from all RSPO constructs were adjusted to equal concentration by western blot and AP activity measurement, and further validated by WNT reporter assay using HEK293T cells. The extracellular domain of BMPR1A (BMPR1A$^{ECD}$) was subcloned in AP-pCS2+ for generating conditioned medium and used in in vitro binding assays. Constitutively active forms of ACVR1, BMPR1A, and BMPR1B (QD) were generated by Gln-Asp mutations as described[60]. HA-tagged BMPR1A/ALK3 was a gift from Dr. D. Koinuma[61].

For *Xenopus* mRNA microinjection, *Xenopus laevis* Bmp4-pCS2+, Rspo2$^{\Delta C}$-myc-pCS2+, Rspo2$^{\Delta FU1/2}$-myc-pCS2+ and Rspo2$^{\Delta TSP}$-myc-pCS2+ plasmids, Bmpr1a$^{DN}$-pCS2+, membrane-RFP, Bmpr1a-EYFP-pCS2+, Rspo2$^{\Delta FU1}$-myc-pCS2+, Rspo2$^{F107E}$-myc-pCS2+ were used for in vitro transcription. Human Noggin-AP-pCS2+ and Chordin-AP-pCS2+ plasmids were used for *Xenopus tropicalis* Crispant rescue assay. Human ZNRF3 and ZNRF3$^{\Delta RING}$ constructs were gifts from Dr. F. Cong (Novartis)[26], and ORFs were further subcloned in flag-pCS2+ for in vitro transcription.

**Cell transfection.** For HEPG2 and H1581 cells, siRNAs and plasmids were transfected using DharmaFECT 1 transfection reagent (Dharmacon T-2001) and Lipofectamine 3000 (Invitrogen L3000), respectively, according to the manufacturer protocols. For HEK293T cells, X-tremeGENE 9 DNA transfection reagent (Roche 6365787001) was used, according to the manufacturer protocols.

**Generation of conditioned medium.** HEK293T cells were seeded in 15 cm culture dishes and transiently transfected with RSPOs-AP, RSPOs-flag, BMPR1A$^{ECD}$-AP, DKK1 or WNT surrogate plasmids. After 24 h, media were changed with fresh DMEM, 10% FBS, 1% L-glutamine and 1% penicillin-streptomycin and cultured 6 days at 32 °C. Conditioned media were harvested three times every two days, centrifuged and validated by TOPFlash assay or western blot analyses. Mouse WNT3A conditioned medium was produced from mouse L-cells stably transfected with WNT3A (ATCC CRL-2647) following the manufacturer's instruction. For human RSPO2$^{\Delta FU1}$, RSPO2$^{\Delta FU2}$, *Xenopus* Rspo2$^{\Delta FU1}$, and Rspo2$^{F107E}$ mutants conditioned media, HEK293T cells were seeded in 12 well culture plates and transfected with 500 μg of each plasmid, and harvested three times every two days. Production of the media was validated with western blot analyses and AP activity analyses.

**Luciferase reporter assays**. BRE luciferase assays were executed using 300,000 ml$^{-1}$ of HEPG2 cells in 24-well plates. PGL3-BRE-Luciferase (500 ng ml$^{-1}$) and pRL-TK-Renilla plasmids (50 ng ml$^{-1}$) were transfected using Lipofectamine 3000. After 24 h, cells were serum starved 2 h and stimulated 14–16 h with 80 ng ml$^{-1}$ recombinant human BMP4 protein (R&D systems 314-BP) along with RSPO1-4 conditioned medium. TOPFlash luciferase assays were executed in HEK293T cells using 10,000 cells per well in 96-well plates. TOPFlash-Luciferase (5 ng) and pRL-TK-Renilla plasmids (1 ng) per well were transfected using X-tremeGENE9 transfection reagent (Roche 06365787001). After 48 h, cells were stimulated 24 h with WNT3A conditioned medium along with RSPO conditioned medium. Luciferase activity was measured with the Dual luciferase reporter assay system (Promega E1960). Firefly luminescence (BRE or TOPFlash) was normalized to Renilla. Statistical analyses were made with Graphpad PRISM7 software.

**Western blot analysis**. Cultured cells were rinsed with cold PBS and lysed in Triton lysis buffer (20 mN Tris-Cl, pH 7.5, 1% Triton X-100, 150 mM NaCl, 1 mM EDTA, 1 mM EGTA, 1 mM β-glycerophosphate, 1 mM Na$_3$VO$_4$) or RIPA buffer with cOmplete Protease Inhibitor Cocktail (Roche 11697498001). Lysates were mixed with Laemmli buffer containing β-mercaptoethanol and boiled at 95 °C for 5 min to prepare SDS-PAGE samples. Western blot images were acquired with SuperSignal West pico ECL (ThermoFisher 34580) or Clarity Western ECL (Biorad 1705061) using LAS-3000 system (FujiFilm). Quantification of blots was done using Image J v1.51k software.

**Cell surface biotinylation assay**. H1581 cells were seeded in 6 cm culture dishes and transfected with 50 nM of indicated siRNAs for 3 days and 2 μg of BMPR1A-HA DNA for 2 days. Surface proteins were biotinylated with 0.25 mg ml$^{-1}$ sulfo-NHS-LC-LC-Biotin (ThermoFisher 21338) at 4 °C for 30 min. The reaction was quenched by 10 mM Monoethanolamine and cells were harvested and lysed with Triton X-100 lysis buffer. 200–300 μg of lysate was incubated with 20 μl streptavidin agarose (ThermoFisher 20359) to pull-down biotinylated surface proteins and subjected to Western blot.

**Surface receptor internalization assay**. H1581 cells were seeded in 15 cm culture dish and transfected with 10 μg of BMPR1A-HA DNA for 2 days, and then split to 6 cm culture dishes. After 24 h, surface proteins were biotinylated with 0.5 mg ml$^{-1}$ sulfo-NHS-SS-Biotin (ThermoFisher 21331) at 4 °C for 30 min. After quenching excessive biotin with 10 mM monoethanolamine, pre-warmed control medium or RSPO2 conditioned medium was added at 37 °C to induce internalization. After 20 min stimulation, remaining surface-biotin was removed by 50 mM MesNa (2-mercaptoethanesulfonate, membrane impermeable reducing agent, CAYMAN 21238) in MesNa reaction buffer (100 mM Tris-HCl, pH 8.6, 100 mM NaCl and 2.5 mM CaCl$_2$) at 4 °C for 30 min and MesNa protected-biotinylated proteins (internalized proteins) were analyzed. Cells were harvested, and lysed with RIPA buffer (20 mM Tris-Cl, pH 7.4, 120 mM NaCl, 1% Triton X-100, 0.25% Na-deoxycholate, 0.05% SDS, 50 mM sodium fluoride, 5 mM EDTA, 2 mM Na-orthovanadate) supplemented with complete protease inhibitor. Five hundred microgram of lysate was incubated with 20 μl streptavidin agarose (ThermoFisher 20359) to pull-down biotinylated proteins and subjected to Western blot.

**Xenopus laevis whole-mount in situ hybridization**. Whole-mount in situ hybridizations of Xenopus embryos were performed using digoxigenin (DIG)-labeled probes according to the standard protocol (https://www.xenbase.org)[62]. Antisense RNA probes against rspo2 and bmp4 were generated by in vitro transcription with T7 RNA polymerase (Promega P2075)[20]. Probes against bmpr1a and znrf3 were prepared using full-size Xenopus bmpr1a ORF or znrf3 ORF as a template, linearized with Xho I (NEB R0146S) and transcribed with T7 RNA polymerase. Mo and mRNA injected embryos were collected at stage 11 (gastrula) or 32 (tadpole) for in situ hybridization. Images were obtained using AxioCam MRc 5 microscope (Zeiss). Embryos in each image were selected using Magnetic Lasso tool or Magic Wand tool of Adobe Photoshop CS6 software, and pasted into the uniform background color for presentation.

**Xenopus microinjection and phenotype analysis**. In vitro fertilization, microinjection and culture of Xenopus embryos were performed according to the standard protocol (https://www.xenbase.org). X.laevis embryos were microinjected with reporter DNAs, in vitro transcribed mRNAs or antisense morpholino oligonucleotide (Mo) using Harvard Apparatus microinjection system. Mos for rspo2[20], lrp6[39], chordin, bmp4[38], znrf3[43] and standard control were purchased from GeneTools. rspo2$^{\Delta TSP}$ Mo was designed based on rspo2 sequence (Supplementary Table 3). X.laevis 4-cell stage embryos were microinjected 5 nl per each blastomere equatorially and cultured until indicated stages. Equal amount of total mRNA or Mo were injected by adjustment with ppl or standard control Mo. Scoring of phenotypes was executed blind from two individuals, and data are representative images from at least two independent experiments. Embryos in each image were selected using Magnetic Lasso tool or Magic Wand tool of Adobe Photoshop CS6 software, and pasted into the uniform background color for presentation.

**Xenopus tropicalis CRISPR/Cas9-mediated mutagenesis**. The 5′ region of genomic sequences from X.tropicalis chordin (NM_001142657.1) and noggin (NM_001171898.1) were searched for guide RNA (gRNA) targeting sites using an online prediction tool (https://crispr.cos.uni-heidelberg.de). Primers for rspo2, noggin and chordin were designed or chosen[27] for PCR-based gRNA template assembly (Supplementary Table 4)[63]. A primer lacking any target sequences was used as control gRNA. PCR reactions were performed with Phusion Hot Start Flex DNA Polymerase (NEB M0535), followed by in vitro transcription using MEGAscript T7 Transcription Kit (Invitrogen AM1334). Embryos were microinjected at one to two-cell stages with a mixture of 50 pg of gRNA and 1 ng of recombinant Cas9 protein (Toolgen) per embryo. Injected embryos were cultured until stage 30, fixed with MEMFA for phenotypical analysis. Scoring of phenotypes was executed at stage 30 with blinding from two individuals, and data are representative images from three independent experiments. Defects were categorized by the severity of ventralization. 'Severe' showed small head, enlarged ventral tissues and short body axis. 'Mild' showed one or two of the defects described above. 'Normal' showed no visible differences to the uninjected control.

**Injected amount of reagents per Xenopus embryo**. Equal amounts of total RNA or Mo were injected by adjustment with preprolactin (PPL) mRNA, control gRNA or standard control Mo. Per embryo; Fig. 2b, 250 pg of bmp4, rspo2 and rspo2 mutants mRNA; Fig. 2c, 5 ng or 10 ng of rspo2 Mo and 5 ng of lrp6 Mo, 300 pg of reporter DNA; Fig. 2d, 15 ng of bmp4 Mo, 2, 5, or 10 ng of rspo2 Mo, 300 pg of reporter DNA; Fig. 2f, 15 ng of bmp4 Mo and 5 ng of rspo2 Mo; Fig. 2h, 50 pg of gRNA, 200 pg of bmp4 mRNA, 2 ng of lrp6 Mo, 1 ng of Cas9 protein; Fig. 3c, e, 500 pg of bmp4; Fig. 5b, c, 20 ng of rspo2 Mo and rspo2$^{\Delta TSP}$ Mo, 300 pg of reporter DNA; Fig. 5e, g, 8 ng of chd Mo, 20 ng of rspo2$^{\Delta TSP}$ Mo and 200 pg of bmpr1a$^{DN}$; Fig. 6c, 500 pg of bmp4 and bmpr1a-EYFP, 250 pg of membrane-RFP, rspo2, and rspo2 deletion mutants mRNA; Fig. 6e, 500 pg of bmp4 and bmpr1a-EYFP, 250 pg of rspo2, and rspo2 deletion mutants, and gfp; Fig. 6g, i, 10 ng of rspo2 Mo and rspo2$^{\Delta TSP}$ Mo; Fig. 7c, 40 ng of znrf3 Mo, 100 pg and 200 pg of rspo2 mRNA, and 300 pg of reporter DNA; Fig. 7d, 500 pg of bmp4 and bmpr1a-EYFP, 250 pg of membrane-RFP, 250 pg of rspo2, 100 pg of znrf3$^{DN}$ mRNA; Fig. 7h, 500 pg of bmp4 and bmpr1a-EYFP, 250 pg of membrane-RFP, rspo2, and rspo2 mutants mRNA; Supplementary Fig. 2b, 15 ng of bmp4 Mo and 5 ng of rspo2 Mo; Supplementary Fig. 3g, i, 50 pg of gRNA, 10 pg and 25 pg of chordin and noggin DNA; Supplementary Fig. 4b, d, f, 50 pg of gRNA, 200 pg of bmp4 mRNA, 2 ng of lrp6 Mo, 1 ng of Cas9 protein; Supplementary Fig. 4h, 250 pg of bmp4 and rspo3 mRNA; Supplementary Fig. 6b, 20 ng of rspo2$^{\Delta TSP}$ Mo, 150 ng and 250 ng of rspo2 mRNA; Supplementary Fig. 6d, 15 ng chd Mo, 10 ng rspo2 Mo and 50 pg bmpr1a$^{DN}$; Supplementary Fig. 7e, 80 ng of znrf3 Mo, 200 pg of ZNRF3 mRNA; Supplementary Fig. 10c, 500 pg of bmpr1a-EYFP.

**Xenopus tropicalis T7 Endonuclease I assay**. To validate CRISPR/Cas9-mediated genome editing, three embryos of each injection set were lysed at stage 30 for genotyping PCR reactions as described[63]. For primer sequences, see Supplementary Table 4. All target sequences were amplified with Roti-Pol Hot-TaqS Mix (Roth 9248). After denaturation for 3 min at 94 °C and reannealing (ramp 0.1 °C per sec), the PCR products were incubated with 3 U of T7 Endonuclease I for 45 min at 37 °C. Cleavage results were visualized on a 2% agarose gel.

**Xenopus laevis western blot analysis**. Injected Xenopus embryos were harvested at stage 15 to 18, homogenized in NP-40 lysis buffer (2% NP-40, 20 mM Tris-HCl pH 7.5, 150 mM NaCl, 10 mM NaF, 10 mM Na$_3$VO$_4$, 10 mM sodium pyrophosphate, 5 mM EDTA, 1 mM EGTA, 1 mM PMSF, and cOmplete Protease Inhibitor Cocktail) with a volume of 20 μl per embryo. Lysates were cleared with CFC-113 (Honeywell 34874), followed by centrifugation (18,800 × g, 10 min at 4 °C), boiling at 95 °C for 5 min with NuPAGE Sample Buffer. 0.5–1 embryos per lane were loaded for SDS-PAGE analysis.

**Cycloheximide treatment on Xenopus laevis animal cap explants**. Xenopus laevis animal caps were dissected at stage 8 and treated with 30 μg ml$^{-1}$ of cycloheximide (CHX) (Sigma C7698) until control embryos reached stage 10. CHX treatment was validated since cell division was retarded compared to untreated control. In situ hybridization and qRT-PCR were performed with same methods used in whole embryos.

**In vitro binding assay**. High binding 96-well plates (Greiner M5811) were coated with 2 μg ml$^{-1}$ of recombinant human RSPO1 (Peprotech 120-38), RSPO2 (Peprotech 120-43), RSPO3 (Peprotech 120-44), RSPO4 (R&D systems 4575-RS) or FGF8b (Peprotech 100-25) recombinant protein reconstituted in bicarbonate coating buffer (50 mM NaHCO$_3$, pH 9.6) overnight at 4 °C. Coated wells were washed three times with TBST (TBS, 0.1% Tween-20) and blocked with 5% BSA in TBST for 1 h at room temperature. 1.5 U ml$^{-1}$ of BMPR1A$^{ECD}$-AP or control conditioned medium was incubated overnight at 4 °C. Wells were washed six times with TBST and bound AP activity was measured by the chemiluminescent SEAP Reporter Gene Assay kit (Abcam ab133077) or AquaSpark AP substrate (Serva 42593.01). For ZNRF3-BMPR1A binding assay, plates were coated with recombinant human ZNRF3 Fc Chimera protein (R&D systems 7994-RF). RSPO2-flag,

RSPO2 deletion mutants-flag conditioned medium, or recombinant RSPO protein was preincubated 4–6 h with ZNRF3 prior to BMPR1A$^{ECD}$-AP treatment. Control conditioned medium and vesicles were used as control. Data show average chemiluminescent activities with SD from experimental triplicates. Statistical analyses show unpaired $t$-tests. The $K_d$ was obtained with In vitro binding assay using RSPO2 recombinant protein (Peprotech 120-43) and BMPR1A$^{ECD}$-AP[24].

**Immunofluorescence**. 150,000 H1581 cells were grown on coverslips in 12-well plates, followed by siRNA and DNA transfection. After 48 h cells were fixed in 4% PFA for 10 min. Cells were treated with primary antibodies (1:250) overnight at 4 °C, and secondary antibodies (1:500) and Hoechst dye (1:500) were applied for 2 h at room temperature. For Tyramide Signal Amplification to detect RSPOs, H1581 cells were treated with RSPO1-3$^{ΔC}$-HRP conditioned media 3 h at 37 °C. Cells were washed with Hanks' balanced salt solution (HBSS) and fixed for 30 min with 0.5 mM dithiobis (succinimidyl) propionate (DSP) (Thermo 22585) crosslinker in HBSS supplemented with 10 mM HEPES, followed by permeabilization with 0.1% saponin in TSA buffer (100 mM Tris, pH 8.8, 10 mM imidazole). The TSA reaction was executed for 30 min in dark with the 30 μM tyramide-Rhodamine in TSA buffer supplemented with 0.003% $H_2O_2$. Cells were washed and further stained with anti-HA and anti-Flag antibodies (1:250)[24,39]. Quantification was done using Image J v1.51k software.

For *X. laevis* embryos, *bmpr1a*-EYFP and membrane-RFP mRNAs were coinjected with the indicated mRNAs or Mos. Embryos were dissected for animal or ventrolateral explants at stage 9 or stage 11.5, respectively. Explants were immediately fixed with 4% PFA for 2 h and mounted with Fluoromount-G (ThermoFisher 00495802). Images were obtained using LSM 700 (Zeiss). Data are representative images from two independent experiments. For quantification, Pearson's correlation coefficient for EYFP and RFP was analyzed using 16–30 random areas harboring 10 cells chosen from 6–10 embryos per each set.

**Cell surface binding assay**. Two hundred and fifty nanogram of human BMPR1A-HA and *Xenopus tropicalis* LGR4 DNA were transfected in HEK293T cells and incubated with 1.5 U ml$^{-1}$ conditioned media for 3 h on ice. After several washes with PBS and crosslinking with DSP, cells were treated with 2 mM Levamisole for 20 min to inactivate endogenous AP activities and developed with BM-Purple (Sigma 11442074001). Cells were mounted with Fluoromount G. Images were obtained using LEICA DMIL microscope/Canon DS126311 camera.

**Quantitative real-time PCR**. Cultured cells were lysed in Macherey-Nagel RA1 buffer containing 1% β-mercaptoethanol and total RNAs were isolated using NucleoSpin RNA isolation kit (Macherey-Nagel 740955). Reverse transcription and PCR amplification were performed as described before[64]. For *Xenopus laevis*, animal cap explants were harvested at stage 10 and lysed in 1 ml of TRizol (Ambion 18914101). Extraction and precipitation of RNA were performed following the manufacturer's instruction. Reverse transcription was executed with 1 μg RNA using SuperScript II reverse transcriptase and random primers (Invitrogen P/N58875). PCR amplification with the obtained cDNA was performed using UPL (Universal Probe Library, Roche, #15 (*rspo2*), #140 (*sizzled*), #145 (*vent1*)) probes and corresponding primers, further analyzed by LightCycle 480. Primers used in this study are listed in Supplementary Table 1. Graphs show relative gene expressions to GAPDH. Data are displayed as mean with SD from multiple experimental replicates. Statistical analyses were performed using the GraphPad PRISM7 software.

**Statistics and reproducibility**. All exact $P$ values in the analyses are as follows. (Left to right of the graph) Fig. 1; (**a**), <0.0001, 0.0007, <0.0001, 0.0008; (**b**), <0.0001, 0.0007, 0.0002, 0.003, 0.0013; (**e**), 0.0064, 0.0009; (**f**), 0.0012, 0.0005; (**h**), 0.0006, <0.0001; (**j**), 0.0347, 0.0132, 0.0040, 0.1778, 0.9918, 0.0541, 0.9925, 0.9082, 0.3433, 0.9920, 0.9991, 0.8893. Figure 2; (**c**), 0.0219, 0.1122, 0.0407, 0.0259; (**d**), 0.0153, 0.0458, 0.0233, 0.0001, 0.0096; (**h**), 0.0007, 0.0650, <0.0001, 0.0035. Figure 3; (**c**), 0.0039, 0.0085, <0.0001, 0.7836, 0.0036, 0.3596. Figure 4; (**a**), 0.0013, 0.0020, 0.0007; (**b**), 0.1193, 0.2732, 0.2591, 0.1096, 0.7246, 0.6867, 0.8089, 0.5742; (**c**), 0.9290, 0.9059, 0.1857, 0.0216, 0.4426, 0.9921, 0.9994, 0.9995; (**e**), 0.6542, 0.0007, 0.0033, 0.7650, 0.2872; (**h**), 0.0047; (**j**), 0.0006, 0.2410, 0.8921; (**k**), 0.0044, 0.0005, 0.0036. Figure 5; (**b**), 0.0766, <0.0001; (**c**), <0.0001, 0.0002; (**e**), 0.0006, <0.0001, 0.0005; (**g**), <0.0001, 0.0019, <0.0001, <0.0001, 0.0001. Figure 6; (**e**), <0.0001, 0.1072, 0.0523; (**h**), 0.0020, 0.0005. Figure 7; (**a**), 0.0011, 0.0104, 0.0043, 0.0031, 0.0001, 0.9979, 0.9973, 0.9997, 0.9938, 0.2378, 0.2508, 0.9775, 0.9894; (**b**), 0.6982, 0.0614, 0.0034, 0.0030; (**c**), <0.0001, 0.1144; (**e**), <0.0001, 0.5172, <0.0001; (**f**), 0.0188, 0.0005, 0.8546, 0.9962; (**g**), 0.0001, 0.0003, 0.0016, 0.0465, 0.0014; (**i**), <0.0001, 0.5588, <0.0001, <0.0001. Figure 8; (**a**), <0.0001, <0.0001, <0.0001; (**b**), 0.0002, 0.0026, 0.0004; (**i**), <0.0001, 0.0643, 0.8264; (**m**), 0.0011, 0.0015; (**o**), 0.0137, 0.0004, 0.2874, 0.1482; (**q**), <0.0001; (**s**), <0.0001. For all $P$ values of Supplementary Figures, see Supplementary Information.

**Reporting summary**. Further information on research design is available in the Nature Research Reporting Summary linked to this article.

## Data availability

NCBI Reference Sequences were used for *Xenopus tropicalis* chordin: [https://www.ncbi.nlm.nih.gov/nuccore/NM_001142657.1]; *Xenopus tropicalis* noggin: [https://www.ncbi.nlm.nih.gov/nuccore/NM_001171898.1]. All raw images of our blotting and all relevant data generated and analyzed in this study are provided in the Source Data file. Source data are provided with this paper.

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

## Acknowledgements

We thank F. Cong for providing the ZNRF3 constructs; D. Koinuma for providing the ALK constructs; C. Janda for providing the WNT surrogate construct; R. Thomas for H1581 cells. We acknowledge G. Roth and Aska Pharmaceuticals Tokyo for generously providing hCG. We thank NXR (RRID: SCR_013731), Xenbase (RRID: SCR_004337), and EXRC for *Xenopus* resources. We thank Fabio da Silva for critical reading of the manuscript. Expert technical support by the DKFZ core facility for light microscopy and the central animal laboratory of DKFZ is gratefully acknowledged. This work was funded by the Deutsche Forschungsgemeinschaft (DFG, German Research Foundation) – SFB1324 – project number 331351713.

## Author contributions

H.L. designed and conducted in vitro, human cell line and *Xenopus* experiments. C.S. conducted human cell line and *Xenopus* experiments. R.S. designed and carried out in vitro and human cell line experiments. A.G. generated materials for the study. H.L. generated illustrations for schematics and models used in the study. All authors analyzed and discussed the data. C.N. conceived and coordinated the study and wrote the paper with contribution from H.L.

## Funding

## Competing interests

The authors declare no competing interests.
