## [Peer Review File · Nature Communications]

REVIEWER COMMENTS

Reviewer #1 (Remarks to the Author):

In this paper entitled "R-spondins are BMP receptor antagonists in early embryonic development", Lee et al., describe the first biochemical and in vivo evidence that RSPO2 (and RSPO3), which are believed to be dedicated WNT enhancers, can also act as direct BMP antagonists, in particular in the context of *Xenopus* early embryonic development. The conclusions put forward in this study challenge the status quo. Because of their significance and likely strong impact of the field, the authors should provide additional assurances that this is solid and readily embraced by the larger WNT and BMP community.

Major questions:

If RSPO2 is part of the BMP4 synexpression group in early *Xenopus* embryos, one would think that in completely ventralized embryos (either by UV irradiation or B-Cat MO-mediated depletion) the levels of endogenous RSPO2 should be very high and circumferential by stage 10. In this case why does high RSPO2 expression not suffice to dorsalize the embryos by inhibiting BMP receptor(s). Is it like Sizzled, which needs an organizer-derived signal namely Chordin to exert its anti-BMP activity? This important question led us to suggest the following critical experiments to prove that RSPO2 indeed inhibits BMP signalling directly at the receptor level without the need of auxiliary secreted factor by the Spemann organizer.

In order to entirely remove all canonical Wnt signalling (and Spemann-derived factors) before the MBT, the authors should inject β -catenin MO in all 4 blastomeres to obtain completely ventralized embryos which will be Sizzledhigh and Sox2- because of ubiquitous and unrestricted BMP2/4/7 signalling. Injection of synthetic mRNA for RSPO2 or RSPO3 should be targeted to the prospective dorsal side and the reappearance of a rescued DV axis (marked by Sox2+ and Sizzledlow) would truly indicate that indeed exogenous RSPO2/3, in the absence of functional WNT and other organizer-derived factors, inhibits endogenous BMP signalling to impart DV polarity. Injections of a DN-ALK3 construct would serve as an ideal positive-control while injections of a construct encoding a WNT ligand should be without any effect.

A similar experiment whereby recombinant RSPO2 or RSPO3 protein is injected into the blastocoel cavity of β -catenin morphants at the blastula stage should also achieve the same type of DV rescue.

These simple yet diagnostic assays harness the power of *Xenopus* and should be employed to their full extent. Because of the numerous feedback loops that exist between BMP and WNT signalling pathways during early development, one has to be extremely careful to avoid the confounding issues associated with the reported transcriptional or post-translational crosstalks. If shown to be positive these experiments would significantly strengthen the conclusions.

While the authors first show that both RSPO2 and RSPO3, but not RSPO1 and RSPO4, decrease BMP4 signaling in HEPG2 cells (Figure 1a), it is not clear why they subsequently choose to only focus on RSPO2 for most of the following experiments, especially in vitro. It would be interesting to check if the same results would have been obtained with RSPO3 (but not RSPO1 and RSPO4).

While the authors use RSPO2 deletion mutants to assess the importance of the different domains (Fu1-Fu2 and TSP1) both in vitro and in vivo, they should use these assays to test the importance of the FU1 and FU2 domains separately. One would expect that only the FU1 domain - which is required for RNF43/ZNRF3 interaction, but not the FU2 domain - which is important for interaction with the LGR4/5/6 is required for antagonizing BMP signaling.

Figure 2b,c shows that *rspo2* overexpression can rescue the ventralization phenotype due to *bmp4* overexpression. While the authors perform the opposite experiment that shows that *rspo2* downregulation by morpholino injection can increase the expression of BMP target genes (Figure 2d-g), they do not show the phenotype of such embryos. Can the injection of *rspo2* MO rescue the dorsalization phenotype of *bmp4* morphants too?

The authors write that "RSPO2 and -3 treatment specifically inhibited ALK3QD but not ALK2QD or ALK6QD (Figure 3a), while RSPO1 and -4 had no effect (Figure 3b)". While this is true for ALK3QD, only the effect of RSPO2 is tested on the other two receptors (Figure 3a). What are the effects of RSPO1-3-4 on ALK2QD and ALK6QD?

The authors then write that "in vitro binding assays revealed that RSPO2 bound to the extracellular domain of ALK3 (Figure 3c) with high affinity (Figure 3d)". However, Figure 3c shows that not only RSPO2 but also RSPO3 (with what Kd?), but not RSPO1 (what about RSPO4?), bind to ALK3.

It would be very compelling to show direct binding between full-length RSPO2 and the TSP1 of RSPO2 (but not that of RSPO1) with the the ECD of ALK3 using crosslinker such as DSS.

The description of Figure 4 is too short in the text and lacks explanation of the experimental plan. There is no mention of the observation that decreased *alk3*-EYFP expression is obtained with RSPO2 WT but not RSPO2 Δ FU or RSPO2 Δ TSP mutants (Figure 4c-e). The same is true for the *rspo2* Δ tsp morpholino injection (Figure 4f-i) for which the design is only described later for Figure 5d and the results from Figure 4f-i are only mentioned much later in the text. In general, the panels in Figures 3-4-5 should be rearranged to better fit with the flow of the text.

In Figure 5g-j experiments, injection of *rspo2* MO should also be performed in parallel to the *rspo2* Δ tsp MO, as they are expected to lead to the same phenotype. However, in the experiment described in Figure 2f-g, it is only mentioned that *sizzled* is still expressed in *rspo2* MO injected embryos. Is it increased? Moreover, with the *rspo2* Crispr-cas9 KO, *sizzled* increase is not as significant as with the *rspo2* Δ tsp morpholino (Figure S2f-g). How do the authors explain this?

The *bf1/myoD/sizzled* triple in situ in stage 32 embryos is very informative in describing the phenotype (Figure 5i). It should definitely be also employed in experiments presented in Figure 2 and Figure S2 to describe more accurately the severity / rescue of the observed phenotype.

The description of Figure 6 is short (2 sentences) and not completely accurate. The authors state that expression of a dominant negative ZNRF3 prevented inhibition of BMP signaling by RSPO2. However, RSPO2 can still reduce, to some extent, BMP signaling and ALK3 stability in presence of ZNRF3 Δ R (Figure 6e-g). Also, how do the authors explain that BMP signaling is increased when *rspo2* mRNA is injected in *znrf3* morphants (Figure 6d)?

The "RSPO2 bridges ALK3 and ZNRF3 and triggers BMP receptor degradation" section should be divided in several sections and expanded, as it describes results from 5 different figures while the first 2 figures have their own sections.

The western blots presented in Figure 7k and 7l are not convincing by themselves. Cell surface ALK3-HA does not seem increased in siZNRF3/RNF43 cells (Figure 7l). The authors should show the localization of ALK3-HA by IF in RNF43/ZNRF3-siRNA treated cells. Does it localize to the cell membrane as in siRSPO2 cells? This is a key missing experiment.

All in all, this is a thought-provoking study which will leave no one indifferent in the BMP and WNT community.

We realize that we have asked many questions and additional experiments.

This reflects our genuine interest in this new concept which if shown to be true should perhaps be

published in sister journal Nature.

Minor points:

The following sentence is misleading: "Interestingly, treatment with RSPO2 and RSPO3 but not RSPO1 and RSPO4 decreased BMP4 signaling, while all RSPOs showed similar ability to amplify Wnt signaling". Indeed, while the BMP signaling analysis (BRE assay) is performed on HEPG2 cells (Figure 1a), the WNT signaling analysis (TOPFLASH assay) is performed on HEK29T cells (Figure S1b). Could the authors perform the SUPERTOPFLASH assay on HEPG2 cells to show that all four RSPOs can induce WNT signaling in HEPG2 cells? This would also be usefully to verify the effect of WNTSG (Fig 1h) and WNT3A (Figure S1h) on WNT signaling in HEPG2 cells.

The following sentence: "Additionally, treatment with the WNT antagonist DKK1 had no effect on BMP signaling (Figure S1k)" is misleading as it comes just after the knockdown experiments performed in H1581 cells. However, the DKK1 experiment was performed in HEPG2 cells. This sentence / result should thus better be placed earlier, after Figure 1h.

The color codes in graphs are often not indicated. Colors should either be removed or indicated in the legend. In general, the figure legends are too short and missing important information.

Loading controls are missing for the western blots in Figure S1d,e,g and j.

Do panels d and e in Figure 2 correspond to BMP-reporter assays (BRE luciferase?) as indicated in the text and figure legends, or to a vent2 qPCR as indicated in the Figure itself?

While the authors write that *rspo2*, *chd* and *noggin* ablation in *Xenopus* "increased BMP target genes", in situ hybridization of only one target gene, *sizzled*, is performed, and this only shows a mild increase (for *rspo2* KO and *chd* KO but nothing is shown for *noggin* KO) (Figure S2f,g). Are the differences statistically significant (Figure S2g)? Could the authors confirm these results with a vent1 in situ and a BMP-reporter assay/vent2 qPCR as performed for the morpholino experiments? In situ hybridization staining is not a quantitative assay so this would require analysis of more BMP target genes for the conclusions to be solid. Also, how are the different categories (normal, mild or severe increase, decrease) established? Is it an expansion of the signal towards the dorsal side? This could be measured more precisely. The levels of *sizzled* expression in the two control embryos shown in Figure 2f and Figure S2f are already very different.

For the phenotype of CRISPR-Cas9 injected embryos, the authors should show how the normal, mild and severe categories are defined. Representative images for *Irp6* MO and *bmp4* mRNA injected embryos are shown in Figure 2h, but no quantification is shown in Figure 2i.

The *rspo2* Δ tsp morpholino (sitting at the exon 4 acceptor site) is described by the authors as generating a *rspo2* mRNA lacking exon 4 and thus leading to a protein lacking the TSP domain (Figure 5d). However, the authors do not show any data supporting this, this morpholino injection could lead to several other splicing defects and nonsense-mediated decay of the mRNA. RT-PCR and RT-qPCR are needed to show the specificity and efficiency of this morpholino on the *rspo2* mRNA.

The authors should provide information on the efficiency of the RSPO2 siRNA used in Figure 4j.

Figure 6b qPCR lacks replicates.

RSPO2 Δ FU1-FU2, RSPO2 Δ FU1 and RSPO2 Δ FU2 should be added to the ZNRF3-ALK3 binding assay described in Figure 7a.

Reviewer #2 (Remarks to the Author):

This is another example of the high quality work to come from the Niehrs lab. It shows the interesting dual function of some of the R-spondins. The one that is already well characterized is the effect on Wnt signaling. Wnt/beta-catenin signaling is attenuated by ZNRF mediated degradation of the LRP receptors, but ZNRF is itself degraded by binding to an RSpO and Lgr5 complex. In this manuscript, R-spondins are shown to also block BMP signaling, and do so by interacting with Alk3/ZNRF, leading to degradation of this specific BMP receptor. This may provide an explanation for the phenotypes of R-spondin knockouts, where the effect on Wnt signaling does not account for loss of function

phenotypes. There is specificity between the Rspos shown, dependent on the TSP domain, and specificity for the BMP receptor, with only Alk3 showing the degradation effect.

The interactions and effect are shown to be biologically meaningful in *Xenopus* embryos, where the Rspo knockdown has a clear effect. Luckily, the major BMP2/4 receptor, Alk3/BMPRIa is the one expressed in early development, with Alk6/Bmpr1b only coming on during gastrulation (Xenbase). This might be noted, since otherwise one would expect some redundancy between receptors.

Extensive interaction and functional tests are used, and so the conclusions are clearly documented.

Minor comments include the observation that the figure legends do not stand by themselves, since there is minimal description of the kinds of experiments done. The authors sentence the reader to have to read the materials and methods if the reader wishes to know what method was used in many cases.

Other

Is the triangle in 1c the right way round?

Legend to figure 3. Terse. Examine materials and Methods? Sad!

Figure 4 Branchial arch, brachial=arm

The official nomenclature for the BMP receptors is a little different from that used throughout the text here. But that is up to the journal style.

Reviewer #3 (Remarks to the Author):

This paper from Christoph Niehrs' group proposes a new role for the R-spondins that have hitherto been described as agonists for the Wnt pathway. The present work shows that RSPO2 and 3, but not RSPO1 and 4 can inhibit BMP signaling, both in the context of tissue culture cells and in *Xenopus* embryos. The authors show that they function by regulating levels of one of the key BMP type I receptors, ALK3, targeting it via the E3 ubiquitin ligase, ZNRF3 for degradation in the lysosome. They also identify the domains of RSPO2 responsible for interaction with ALK3 and with ZNRF3.

On the whole, I think that the work is of high interest to the signaling field and is well done and convincing. I have some comments on controls that are missing and I require some clarifications in terms of how the manuscript is structured. I also think it important for the authors to understand better how RSPO2/3 are functioning physiologically. I.e., whether RSPO2 and 3 purely function by modulating levels of ALK3, or whether their expression/activity is regulated so that RSPO2/3 can integrate other responses with BMP signaling.

1. Concerning the last point above first. The authors show that RSPO2 and 3 function to facilitate the internalization and degradation of ALK3, one of the BMP type I receptors. These data are clear. They also show in *Xenopus* embryos that the expression pattern *rspo2* overlaps with that of *bmp4*. Does BMP signaling regulate *rspo2/3* expression? How are *rspo2/3* regulated. Are they induced by any other signaling pathways?

My question is really about how this negative regulation of *bmp* signaling is regulated in vivo. Do *Rspo2/3* simply dictate the levels of ALK3 at the plasma membrane? If so, how is the extent of the degradation of ALK3 regulated, so that the levels are still sufficient for BMP signaling? Are the levels of *Rspo2/3* regulated by other pathways and thus provide a cross talk between those other pathways and BMP signaling? Also, the type I and type II TGF- β family receptors constantly internalize and recycle to the plasma membrane, thus resulting in a pool at the plasma membrane and an internalized pool. Rather than inducing internalization, are *Rspo2/3* more likely just diverting more of the pool away from recycling and towards the lysosome?

Why is only ALK3 regulated this way? Which is the most dominant BMP type I receptor in *Xenopus* embryos? Is it ALK3, or ALK2 or ALK6?

What happens upon ligand stimulation? Do RSP02/3 compete with BMP4? Which has the higher affinity?

2. For both the MOs and also for the transient CRISPR experiments, it is important to have more controls for off target effects. In both cases, rescues would be the best way to do this.

3. In Figure 2b, the authors show that Rspo2 can rescue defects due to Bmp4 overexpression. The authors need to show what happens when Rspo2 is expressed alone.

4. The authors should explain in the text how the binding assays in Figure 3c and 3e were done. The text is rather cryptic on this point.

5. The structure of the manuscript is a little strange in that reagents such as the R1-TSPR2 are used in earlier figures (Figure 3) and then explained later (with the data for Figure 5). The same is true for the *rspo2* Δ TSP MO. It would be better to either explain them both when they are first used (Figure 3), or wait until the figure 5, but present all the relevant data using them in the same place.

6. In the surface biotinylation assays, why does knockdown of ZNRF3/RNF43 not have much effect?

Response to the Referees

Your manuscript entitled "R-spondins are BMP receptor antagonists in early embryonic development" has now been seen by 3 referees, whose comments are appended below. You will see from their comments copied below that while they find your work of considerable potential interest, they have raised substantial concerns that must be addressed.

Notably, we find it critical that you show Rspo2 can directly inhibit BMP signaling (Reviewer #1), elaborate on the similarities between Rspo2 and Rspo3 with regard to BMP signaling, and perform further experiments to uncover how Rspo2/3 regulate ALK3 levels (Reviewer #3) to be considered for publication.

We appreciate the constructive comments of the editors and Reviewers, which provided valuable guidance to improve our manuscript. We have now addressed the majority of the comments, including by a battery of new experiments, represented in 58 new Figure panels. Notably, regarding the critical concerns the editor emphasized, we now confirm that RSPO2 and -3 share the same mechanism to antagonize BMP signaling (Reviewer #1), and demonstrate Rspo2 as a new negative feedback inhibitor in BMP4 synexpression group, which induces BMPR1A/ALK3 internalization (Reviewer #3). We now present 8 main and 10 supplementary figures. We believe these findings have significantly strengthened our study, opening up re-interpretation of RSPO function in stem cell, development, and cancer biology.

Reviewer 1:

In this paper entitled "R-spondins are BMP receptor antagonists in early embryonic development", Lee et al., describe the first biochemical and in vivo evidence that RSPO2 (and RSPO3), which are believed to be dedicated WNT enhancers, can also act as direct BMP antagonists, in particular in the context of Xenopus early embryonic development. The conclusions put forward in this study challenge the status quo. Because of their significance and likely strong impact of the field, the authors should provide additional assurances that this is solid and readily embraced by the larger WNT and BMP community. All in all, this is a thought-provoking study which will leave no one indifferent in the BMP and WNT community. We realize that we have asked many questions and additional experiments. This reflects our genuine interest in this new concept which if shown to be true should perhaps be published in sister journal Nature.

Major Points

Comment 1. *If RSPO2 is part of the BMP4 synexpression group in early Xenopus embryos, one would think that in completely ventralized embryos (either by UV irradiation or B-Cat MO-mediated depletion) the levels of endogenous RSPO2 should be very high and circumferential by stage 10. In this case why does high RSPO2 expression not suffice to dorsalize the embryos by inhibiting BMP receptor(s). Is it like Sizzled, which needs an organizer-derived signal namely Chordin to exert its anti-BMP activity?*

This important question led us to suggest the following critical experiments to prove that RSPO2 indeed inhibits BMP signalling directly at the receptor level without the need of auxiliary secreted factor by the Spemann organizer.

In order to entirely remove all canonical Wnt signalling (and Spemann-derived factors) before the MBT, the authors should inject β -catenin MO in all 4 blastomeres to obtain completely ventralized embryos which will be Sizzledhigh and Sox2- because of ubiquitous and unrestricted BMP2/4/7 signalling. Injection of synthetic mRNA for RSPO2 or RSPO3 should be targeted to the prospective dorsal side and the reappearance of a rescued DV axis (marked by Sox2+ and Sizzledlow) would truly indicate that indeed exogenous RSPO2/3, in the absence of functional WNT and other organizer-derived factors, inhibits endogenous BMP signalling to impart DV polarity. Injections of a DN-ALK3 construct would serve as an ideal positive-control while injections of a construct encoding a WNT ligand should be without any effect.

A similar experiment whereby recombinant RSPO2 or RSPO3 protein is injected into the blastocoel cavity of β -catenin morphants at the blastula stage should also achieve the same type of DV rescue.

These simple yet diagnostic assays harness the power of Xenopus and should be employed to their full extent. Because of the numerous feedback loops that exist between BMP and WNT signalling pathways during early development, one has to be extremely careful to avoid the confounding issues associated with the reported transcriptional or post-translational crosstalks. If shown to be positive these experiments would significantly strengthen the conclusions.

Regarding the reviewers concern of weak dorsalizing effect of overexpressed Rspo2, the reviewer confuses the mechanism of common BMP antagonists with the mechanism of RSPO2/3, which function specifically as BMPR1A/ALK3 antagonists, not by sequestering BMP ligands. Overexpression of BMP antagonists such as *chordin* and *noggin* completely wipes out BMP signaling by sequestering multiple ligands, thereby strongly dorsalizing embryos and generating axis duplications. In contrast, even high Rspo2 dose will only affect Bmpr1a/Alk3 but

not Bmpr1b/Alk6 and Acvr1/Alk2, which continue to signal and compensate for each other in *Xenopus* (Leibovich *et al.*, 2017; Schille *et al.*, 2016).

Comment 2. *While the authors first show that both RSPO2 and RSPO3, but not RSPO1 and RSPO4, decrease BMP4 signaling in HEPG2 cells (Figure 1a), it is not clear why they subsequently choose to only focus on RSPO2 for most of the following experiments, especially in vitro. It would be interesting to check if the same results would have been obtained with RSPO3 (but not RSPO1 and RSPO4).*

We focused on RSPO2 for economy, expecting that RSPO3 acts similarly. However, we now confirm also for RSPO3-mediated BMP inhibition **(1)** WNT independency, **(2)** destabilization of BMPR1A/ALK3 via the TSP1 domain-BMPR1A/ALK3 binding, **(3)** Requirement of ZNRF3/RNF43, and **(4)** recruitment of ZNRF3 to BMPR1A/ALK3 for ZNRF3-RSPO3-BMPR1A/ALK3 complex formation.

Specifically we now show that among RSPOs, only RSPO2 and -3 decrease phosphorylation of Smad1 (**New Figure 1c-d; New Supplementary Figure 1e-f**), and this effect is independent of WNT/ β -catenin (**New Figure 1a-b**). RSPO3 specifically antagonizes BMPR1A/ALK3^{QD} but not BMPR1B/ALK6^{QD} or ACVR1/ALK2^{QD} signaling (**Figure 4a; New Figure 4b-c**), by way of RSPO3^{TSP1}-BMPR1A/ALK3 binding (**New Supplementary Figure 5d-e**). RSPO3 also requires ZNRF3/RNF43 to antagonize BMP4 signaling (**New Figure 7a**). RSPO3 is internalized together with ZNRF3 and BMPR1A/ALK3, indicating that RSPO3 engages ZNRF3 and BMPR1A/ALK3 like RSPO2 (**New Figure 8d**). Based on these results, we revised our model to include both RSPO2 and -3 (**New Supplementary Figure 10d**).

Comment 3. *While the authors use RSPO2 deletion mutants to assess the importance of the different domains (Fu1-Fu2 and TSP1) both in vitro and in vivo, they should use these assays to test the importance of the FU1 and FU2 domains separately. One would expect that only the FU1 domain - which is required for RNF43/ZNRF3 interaction, but not the FU2 domain - which is important for interaction with the LGR4/5/6 is required for antagonizing BMP signaling.*

To test the importance of the FU1 and FU2 domains separately, we constructed and characterized human RSPO2 FU1- and FU2 deletion mutants (RSPO2 ^{Δ FU1}, RSPO2 ^{Δ FU2}) (**New Supplementary Figure 8a**). Expectedly, deletion of FU1 but not FU2 prevents BMP4 inhibition

(New Figure 7f-g). Moreover, we constructed *Xenopus* Rspo2 FU1 (Rspo2^{ΔFU1}) and Rspo2 FU2 mutants (Rspo2^{F107E}) to analyze each domain *in vivo* (**New Supplementary Figure 8d**). Wildtype Rspo2 and Rspo2^{F107E} destabilize surface Bmpr1a/Alk3 levels in *Xenopus* animal cap tissues, while Rspo2^{ΔFU1} does not (**New Figure 7h-i**). These results confirm that RSPO2 FU1 domain but not FU2 domain is required to antagonize BMP receptor signaling, consistent with the requirement of ZNRF3/RNF43 but not of LGRs.

Comment 4. *Figure 2b,c shows that rspo2 overexpression can rescue the ventralization phenotype due to bmp4 overexpression. While the authors perform the opposite experiment that shows that rspo2 downregulation by morpholino injection can increase the expression of BMP target genes (Figure 2d-g), they do not show the phenotype of such embryos. Can the injection of rspo2 MO rescue the dorsalization phenotype of bmp4 morphants too?*

Yes, this is now shown in **New Supplementary Figure 2a-b**.

Comment 5. *The authors write that “RSPO2 and -3 treatment specifically inhibited ALK3QD but not ALK2QD or ALK6QD (Figure 3a), while RSPO1 and -4 had no effect (Figure 3b)”. While this is true for ALK3QD, only the effect of RSPO2 is tested on the other two receptors (Figure 3a). What are the effects of RSPO1-3-4 on ALK2QD and ALK6QD?*

We performed BRE reporter assay in HEPG2 cells transfected with ACVR1/ALK2^{QD} and BMPR1B/ALK6^{QD} upon all four RSPOs treatment (**New Figure 4b-c**). None of RSPOs affects ACVR1/ALK2 or BMPR1B/ALK6 mediated BMP signaling, supporting our model which suggests RSPO2 and -3 specifically bind and destabilize BMPR1A/ALK3 protein levels.

Comment 6. *The authors then write that “in vitro binding assays revealed that RSPO2 bound to the extracellular domain of ALK3 (Figure 3c) with high affinity (Figure 3d)”. However, Figure 3c shows that not only RSPO2 but also RSPO3 (with what Kd?), but not RSPO1 (what about RSPO4?), bind to ALK3.*

Our new cell surface binding assay (**New Figure 4d**) and *in vitro* binding assay (**New Figure 4e**) now clearly reveal that RSPO4 does not bind to the extracellular domain of BMPR1A/ALK3.

Concerning K_d measurement of other RSPOs- BMPR1A/ALK3^{ECD}, producing, purifying, and characterizing recombinant proteins for three more RSPOs, to carry out K_d measurements including for two RSPOs that will be negative, adds little new information but is a major experimental investment, notably in these challenging times, when we have restricted laboratory access. In view of the many more meaningful other points to be addressed, we hope you agree that we rather focused generating new functional data on RSPO3 (see **Comment 2**).

Comment 7. *It would be very compelling to show direct binding between full-length RSPO2 and the TSP1 of RSPO2 (but not that of RSPO1) with the ECD of ALK3 using crosslinker such as DSS.*

In the cell surface binding assays, we did already crosslink RSPOs with DSP to BMPR1A/ALK3^{ECD} (**Supplementary Figure 5b-c**). In addition, we now added *in vitro* binding assays in **new Figure 4g-h**, which shows that RSPO2^{TSP1}, but not RSPO1^{TSP1}, indeed binds to BMPR1A/ALK3^{ECD}.

Comment 8. *The description of Figure 4 is too short in the text and lacks explanation of the experimental plan. There is no mention of the observation that decreased alk3-EYFP expression is obtained with RSPO2 WT but not RSPO2 Δ FU or RSPO2 Δ TSP mutants (Figure 4c-e).*

We thank the reviewer for constructive suggestion. We now have given the sufficient description of previous Figure 4c-e (now **Figure 6b-e**) accordingly;

*“...Similarly in *Xenopus* whole embryos, microinjection of mRNA encoding *rspo2* but not *rspo2* ^{Δ FU1/2} or *rspo2* ^{Δ TSP} decreased protein levels from coinjected *bmpr1a*-EYFP mRNA (Fig. 6b). Immunofluorescence microscopy (IF) of *Xenopus* animal cap explants showed that *Bmpr1a*-EYFP localizes to the plasma membrane, where it was once again reduced by *rspo2* but not by *rspo2* ^{Δ FU1/2} or *rspo2* ^{Δ TSP} mRNA (Fig. 6c-e).”*

*The same is true for the *rspo2* Δ tsp morpholino injection (Figure 4f-i) for which the design is only described later for Figure 5d and the results from Figure 4f-i are only mentioned much later in the text. In general, the panels in Figures 3-4-5 should be rearranged to better fit with the flow of the text.*

The panels in previous Figures 3-4-5 now have been rearranged in **Figure 4-5-6-7, Supplementary Figure 5, and -7** with new data to improve the flow. As the reviewer pointed out, now *rspo2*^{ΔTSP} Mo is first introduced in **Figure 5**, and previous Figure 4f-i are displayed later as **Figure 6f-i** with sufficient description accordingly;

“...Focusing on *Xenopus ventrolateral marginal zone (VLMZ)* explants, where endogenous *rspo2*, *bmpr1a* and *bmp4* are coexpressed, showed that ablation of *rspo2* by Mo injection results in significant increase of *Bmpr1a*-EYFP plasma membrane levels (Fig. 6f-h). Moreover, in VLMZ from *rspo2*^{ΔTSP} Morphants, *Bmpr1a* levels were also increased (Fig. 6f-h), which was confirmed by western blot analysis (Fig. 6i).”

Comment 9. *In Figure 5g-j experiments, injection of rspo2 MO should also be performed in parallel to the rspo2Δtsp MO, as they are expected to lead to the same phenotype. However, in the experiment described in Figure 2f-g, it is only mentioned that sizzled is still expressed in rspo2 MO injected embryos. Is it increased?*

We now show *in situ* hybridization of *sizzled* and *vent1* with *rspo2* Mo and *bmpr1a/alk3*^{DN} injection, in line with previous Figure 5g-h (**New Supplementary Figure 6c-d**). Similar to *rspo2*^{ΔTSP} Mo, *rspo2* Mo injected gastrulae showed increased *sizzled* and *vent1* expression, and co-injection of *bmpr1a/alk3*^{DN} rescued the increase as the reviewer predicted.

For previous Figure 2f-g (now 2e-f), our purpose was to analyze whether *rspo2* Mo can rescue the reduction of *vent1* and *sizzled* expression caused by *bmp4* Mo. Therefore, we categorized expression patterns to ‘expressed’ or ‘abolished’, not to ‘increased’ or ‘decreased’, to quantify the phenotype.

Moreover, with the rspo2 Crispr-cas9 KO, sizzled increase is not as significant as with the rspo2ΔTSP morpholino (Figure S2f-g). How do the authors explain this?

The two experiments cannot be directly compared. We used *X.tropicalis* for the KO, and *X.laevis* for the Mo injection and Mo and Crispr-cas9 KO affect gene/RNA function very differently.

Comment 10. *The bf1/myoD/sizzled triple in situ in stage 32 embryos is very informative in describing the phenotype (Figure 5i). It should definitely be also employed in experiments*

presented in Figure 2 and Figure S2 to describe more accurately the severity / rescue of the observed phenotype.

In the quantifications in Figure 2 and Figure S2 (now Figure S4) we carefully categorized embryos phenotypically and scored embryos with blinding by two individuals, which is now described in the legends and methods. The conclusion derived from the phenotype analyses was further confirmed in *Xenopus* throughout the MS (Figure 2c-f, Figure 5, Supplementary Figure 2, Supplementary Figure 4).

Comment 11. *The description of Figure 6 is short (2 sentences) and not completely accurate. The authors state that expression of a dominant negative ZNRF3 prevented inhibition of BMP signaling by RSPO2. However, RSPO2 can still reduce, to some extent, BMP signaling and ALK3 stability in presence of ZNRF3^{ΔR} (Figure 6e-g).*

Panels in previous Figure 6 are now displayed in **Figure 7** and **Supplementary Figure 7** with new data, and the results are described in an independent paragraph. For previous Figure 6e, we repeated and improved the result, which shows that RSPO2 does not reduce BMP signaling in presence of ZNRF3^{ΔR} (**New Figure 7b**). For previous Figure 6g, we mislabeled the P value. We apologize for the mistake, which has been corrected in **New Figure 7e**.

*Also, how do the authors explain that BMP signaling is increased when *rspo2* mRNA is injected in *znrf3* Morphants (Figure 6d)?*

We now performed one-way ANOVA test with the existing data to analyze whether there is the difference between *znrf3* Morphants and two *rspo2* mRNA injected groups. There was no significant difference (**New Figure 7c**).

Comment 12. *The “RSPO2 bridges ALK3 and ZNRF3 and triggers BMP receptor degradation” section should be divided in several sections and expanded, as it describes results from 5 different figures while the first 2 figures have their own sections.*

As the reviewer suggested, the whole “RSPO2 bridges ALK3 and ZNRF3 and triggers BMP receptor degradation” section now has been divided into four sections, and supplemented with more detailed descriptions.

Comment 13. *The western blots presented in Figure 7k and 7l are not convincing by themselves. Cell surface ALK3-HA does not seem increased in siZNF3/RNF43 cells (Figure 7l).*

We repeated the surface biotinylation assays with longer siRNA treatment, and improved the data showing now convincing increase of cell surface BMPR1A/ALK3 levels upon knockdown of ZNF3/RNF43 (**New Figure 8n**).

Comment 14. *The authors should show the localization of ALK3-HA by IF in RNF43/ZNF3-siRNA treated cells. Does it localize to the cell membrane as in siRSPO2 cells? This is a key missing experiment.*

Yes, BMPR1A/ALK3-HA localizes to the cell membrane as in siRSPO2 cells. We performed IF for BMPR1A/ALK3-HA localization in siControl, siRSPO2, and siZNF3/RNF43 treated H1581 cells and found that BMPR1A/ALK3-HA is indeed accumulated at the cell membrane upon both siZNF3/RNF43 and siRSPO2 treatment (**New Figure 8j-m**). This result along with the new Figure 8n (**Comment 13**), now clearly demonstrates that loss of ZNF3/RNF43 increases cell surface BMPR1A/ALK3, consistent with our model.

Minor Points

Comment 15. *The following sentence is misleading: “Interestingly, treatment with RSPO2 and RSPO3 but not RSPO1 and RSPO4 decreased BMP4 signaling, while all RSPOs showed similar ability to amplify Wnt signaling”. Indeed, while the BMP signaling analysis (BRE assay) is performed on HEPG2 cells (Figure 1a), the WNT signaling analysis (TOPFLASH assay) is performed on HEK29T cells (Figure S1b). Could the authors perform the SUPERTOPFLASH assay on HEPG2 cells to show that all four RSPOs can induce WNT signaling in HEPG2 cells? This would also be usefully to verify the effect of WNTSG (Fig 1h) and WNT3A (Figure S1h) on WNT signaling in HEPG2 cells.*

It is known that HEPG2 cells have a truncation mutation of the exon 3-4 in *CTNNB1* which encodes β -catenin protein, hence induces constitutive activation of WNT/ β -catenin signaling (de La Coste *et al.*, 1998). Additional RSPO1-4 treatment would not sensitize WNT signaling in

HEPG2 cells. Therefore, we used HEK293T cells for TOPFLASH assays. Note that the knockdown of β -catenin in HEPG2 cells to analyze WNT independent role of RSPO2 is still valid, because si β -catenin yields complete loss of TOPFlash activity with significant reduction of both wild-type and truncated form of β -catenin proteins (**Supplementary Figure 1c-d**).

Comment 16. *The following sentence: “Additionally, treatment with the WNT antagonist DKK1 had no effect on BMP signaling (Figure S1k)” is misleading as it comes just after the knockdown experiments performed in H1581 cells. However, the DKK1 experiment was performed in HEPG2 cells. This sentence / result should thus better be placed earlier, after Figure 1h.*

As the reviewer suggested, previous Supplementary Figure 1k has been placed after previous Supplementary Figure 1h as **Supplementary Figure 1l**.

Comment 17. *The color codes in graphs are often not indicated. Colors should either be removed or indicated in the legend. In general, the figure legends are too short and missing important information.*

We utilized different color bars to help readers to notice decrease or increase easily. Otherwise, color codes are indicated completely when it is necessary. We now have provided the revised figure legends with sufficient information.

Comment 18. *Loading controls are missing for the western blots in Figure S1d,e,g and j.*

Western blots for loading controls are now added to **Supplementary Figure 1d, i, j and o** (previous Figure S1d,e,g and j.).

Comment 19. *Do panels d and e in Figure 2 correspond to BMP-reporter assays (BRE luciferase?) as indicated in the text and figure legends, or to a vent2 qPCR as indicated in the Figure itself?*

They correspond to luciferase assays using the vent2 reporter.

Comment 20. *While the authors write that *rspo2*, *chd* and *noggin* ablation in *Xenopus* “increased BMP target genes”, in situ hybridization of only one target gene, *sizzled*, is performed, and this only shows a mild increase (for *rspo2* KO and *chd* KO but nothing is shown for *noggin* KO) (Figure S2f,g). Are the differences statistically significant (Figure S2g)?*

For *rspo2* KO and *chd* KO, we added new *in situ* hybridization of *vent1* (**New Supplementary Figure 4c-d**). For *rspo2* KO and *nog* KO, we performed *in situ* hybridization of *vent1* and *sizzled* (**New Supplementary Figure 4e-f**). X^2 tests confirmed that the differences for increase of BMP target genes in *rspo2/chd* and *rspo2/nog* DKO compared with each KO are statistically significant, and now indicated in **Supplementary Figure 4d** and **4f**.

Comment 21. *Could the authors confirm these results with a *vent1* in situ and a BMP-reporter assay/*vent2* qPCR as performed for the morpholino experiments? In situ hybridization staining is not a quantitative assay so this would require analysis of more BMP target genes for the conclusions to be solid.*

As in **Comment 20**, new *vent1* *in situs* were added to previous Figure S2g (**New Supplementary Figure 4c-d**), which support our conclusion.

Comment 22. *Also, how are the different categories (normal, mild or severe increase, decrease) established? Is it an expansion of the signal towards the dorsal side? This could be measured more precisely.*

Yes. We categorized *sizzled* and *vent1* expression based on how severely the signal expanded to the dorsal side. Specifically, we measured the central angle of the expression ‘arc’ for every embryos, and classified them into the different categories. For *sizzled*, ‘normal’ showed embryos with the angles of the expression as 40°. ‘mild increase’ showed embryos with the angles between 40°-80° and ‘severe increase’ showed embryos with the angles higher than 80°. For *vent1*, ‘normal’ showed embryos with the angles of the expression as 260°. ‘mild increase’ showed embryos with the angles between 260°-320° and ‘severe increase’ showed embryos with the angles higher than 320°. Scheme for the categorization is now illustrated in **New Supplementary Figure 4d**.

*The levels of *sizzled* expression in the two control embryos shown in Figure 2f and Figure S2f are already very different.*

As in **Comment 9**, different levels of *sizzled* expression comes from the usage of different *Xenopus* species.

Comment 23. *For the phenotype of CRISPR-Cas9 injected embryos, the authors should show how the normal, mild and severe categories are defined. Representative images for *Irp6* MO and *bmp4* mRNA injected embryos are shown in Figure 2h, but no quantification is shown in Figure 2i.*

We had indicated in the figure legend how phenotypes of Crispants were categorized;

‘...‘Severe’ showed small head, enlarged ventral tissues and short body axis. ‘Mild’ showed one or two of the defects described above. ‘Normal’ showed no visible differences to the uninjected control.’

Quantification of *Irp6* Morphants and *bmp4* injected embryos is now shown in **Figure 2h**.

Comment 24. *The *rspo2* Δ *tsp* morpholino (sitting at the exon 4 acceptor site) is described by the authors as generating a *rspo2* mRNA lacking exon 4 and thus leading to a protein lacking the TSP domain (Figure 5d). However, the authors do not show any data supporting this, this morpholino injection could lead to several other splicing defects and nonsense-mediated decay of the mRNA. RT-PCR and RT-qPCR are needed to show the specificity and efficiency of this morpholino on the *rspo2* mRNA.*

Figure 5 shows that *rspo2* Δ *TSP* Mo injection resulted in a BMP-specific phenotype, while it had no effect on WNT signaling, entirely consistent with our FU1, FU2, TSP1 deletion analyses. These results are different from *rspo2* Mo injection which completely ablates endogenous *rspo2* thereby reduces WNT signaling. To further rule out the off-target effect, we performed phenotype rescue assay (**New Supplementary Figure 6a-b**, related to **the reviewer #3 comment 5**). *rspo2* Δ *TSP* Mo yielded ventralized tadpoles, and co-injection of *rspo2* mRNA efficiently rescued this defect.

Comment 25. *The authors should provide information on the efficiency of the RSPO2 siRNA used in Figure 4j.*

The knockdown efficiency of *RSPO2* siRNA was already provided in **Supplementary Figure 1n**, which shows 96% of reduction for *RSPO2* expression. We had kept the same experimental condition for si*RSPO2* treatment throughout the MS.

Comment 26. *Figure 6b qPCR lacks replicates.*

We repeated qPCR, and the new result with triplicates is now shown in **New Supplementary Figure 7a**.

Comment 27. *RSPO2 Δ FU1-FU2, RSPO2 Δ FU1 and RSPO2 Δ FU2 should be added to the ZNRF3-ALK3 binding assay described in Figure 7a.*

As the reviewer suggested, we now demonstrate *in vitro* binding assays for ZNRF3-BMPRI1A/ALK3 upon each *RSPO2* mutant treatment. Wildtype *RSPO2* and *RSPO2* ^{Δ FU2} are able to engage ZNRF3 and BMPRI1A/ALK3, however, *RSPO2* ^{Δ FU1/2} and *RSPO2* ^{Δ FU1} do not mediate ZNRF3-BMPRI1A/ALK3 interaction (**New Supplementary Figure 9c-h**). These results indicate that the FU1- but not the FU2 domain is required for ZNRF3-*RSPO2*-BMPRI1A/ALK3 complex formation, which is consistent with our model.

Referee #2:

This is another example of the high quality work to come from the Niehrs lab. It shows the interesting dual function of some of the R-spondins. The one that is already well characterized is the effect on Wnt signaling. Wnt/beta-catenin signaling is attenuated by ZNRF mediated degradation of the LRP receptors, but ZNRF is itself degraded by binding to an Rspo and Lgr5 complex. In this manuscript, R-spondins are shown to also block BMP signaling, and do so by interacting with Alk3/ZNRF, leading to degradation of this specific BMP receptor. This may provide an explanation for the phenotypes of R-spondin knockouts, where the effect on Wnt signaling does not account for loss of function phenotypes. There is specificity between the R-spos shown, dependent on the TSP domain, and specificity for the BMP receptor, with only Alk3 showing the degradation effect.

Comment 1. *The interactions and effect are shown to be biologically meaningful in Xenopus embryos, where the Rspo knockdown has a clear effect. Luckily, the major BMP2/4 receptor, Alk3/BMPR1a is the one expressed in early development, with Alk6/Bmpr1b only coming on during gastrulation (Xenbase). This might be noted, since otherwise one would expect some redundancy between receptors. Extensive interaction and functional tests are used, and so the conclusions are clearly documented.*

There is in fact partial redundancy between BMP receptors in early embryos, e.g. Bmpr1a/Alk3 and Bmpr1b/Alk6 (Leibovich *et al.*, 2017; Schille *et al.*, 2016) (now added in the Discussion). Indeed, if Bmpr1a/Alk3 was the only relevant BMP receptor in early embryos, *rspo2* mRNA injection should induce very strong dorsalization and secondary axes, but it does not. So, *Rspo2* is very different from BMP antagonists such as *chordin*, which sequester most BMPs and thereby strongly dorsalize. *Rspo2* function resembles more that of *Bambi*, another synexpressed negative feedback BMP4 inhibitor, or *Sef* in the FGF8 synexpression group.

Comment 2. *Minor comments include the observation that the figure legends do not stand by themselves, since there is minimal description of the kinds of experiments done. The authors sentence the reader to have to read the materials and methods if the reader wishes to know what method was used in many cases.*

We apologize for the inconvenience. We extensively revised the figure legends to give sufficient description of the methods.

Comment 3. *Is the triangle in 1c the right way round?*

Previous Figure 1c has been exchanged to new panel, which shows the same result. The triangle in new Figure 1c is presented in the right way.

Comment 4. *Legend to figure 3. Terse. Examine materials and Methods? Sad!*

We apologize for the inconvenience. We extensively revised the figure legends for previous Figure 3 (now in Figure 4 and Supplementary Figure 5).

Comment 5. *Figure 4 Branchial arch, brachial=arm*

We thank the reviewer, it has been corrected.

Comment 6. *The official nomenclature for the BMP receptors is a little different from that used throughout the text here. But that is up to the journal style.*

We revised the nomenclature, ALK3 is now BMPR1A.

Referee #3:

This paper from Christoph Niehrs' group proposes a new role for the R-spondins that have hitherto been described as agonists for the Wnt pathway. The present work shows that RSPO2 and 3, but not RSPO1 and 4 can inhibit BMP signaling, both in the context of tissue culture cells and in Xenopus embryos. The authors show that they function by regulating levels of one of the key BMP type I receptors, ALK3, targeting it via the E3 ubiquitin ligase, ZNRF3 for degradation in the lysosome. They also identify the domains of RSPO2 responsible for interaction with ALK3 and with ZNRF3.

On the whole, I think that the work is of high interest to the signaling field and is well done and convincing. I have some comments on controls that are missing and I require some clarifications in terms of how the manuscript is structured. I also think it important for the authors to understand better how RSPO2/3 are functioning physiologically. I.e., whether RSPO2 and 3 purely function by modulating levels of ALK3, or whether their expression/activity is regulated so that RSPO2/3 can integrate other responses with BMP signaling.

Comment 1. *Concerning the last point above first. The authors show that RSPO2 and 3 function to facilitate the internalization and degradation of ALK3, one of the BMP type I receptors. These data are clear. They also show in Xenopus embryos that the expression pattern *rspo2* overlaps with that of *bmp4*. Does BMP signaling regulate *rspo2/3* expression? How are *rspo2/3* regulated. Are they induced by any other signaling pathways?*

Thank you for bringing up the point of Rspo2 possibly being a negative feedback inhibitor. We now established that Rspo2 is, in fact, a novel member of the BMP4 synexpression group in *Xenopus*. Members of the BMP4 synexpression group are expressed like this growth factor and consists of at least eight members, which all encode positive or negative feedback components of the BMP signaling cascade (e.g. Bambi, Twsg1, Smad8, Vent2). qRT-PCR analysis now shows that *rspo2* is induced by Bmp4, similar to known Bmp4 targets *sizzled* and *vent1* (**New Figure 3c**). This finding was further confirmed by *in situ* hybridization (**New Figure 3d-e**), indicating that *rspo2* is a Bmp4 target gene. We further demonstrated that Bmp4 induces *rspo2* indirectly, since cycloheximide treatment abolishes *bmp4*-mediated *rspo2* induction (**New Figure 3c-e**).

Comment 2. *My question is really about how this negative regulation of bmp signaling is regulated in vivo. Do Rspo2/3 simply dictate the levels of ALK3 at the plasma membrane? If so, how is the extent of the degradation of ALK3 regulated, so that the levels are still sufficient for BMP signaling? Are the levels of Rspo2/3 regulated by other pathways and thus provide a cross talk between those other pathways and BMP signaling?*

Similar to **Comment 1**, we now show that Rspo2 is a new negative feedback inhibitor of the BMP4 synexpression group in *Xenopus* (**New Figure 3**). *rspo2* is an indirect BMP target gene, whose expression may depend on transcription factors of the e.g. Vent or Msx families. As we show, the requirement for Rspo2 in inhibiting BMP signaling manifests most strongly when noggin or chordin are also inhibited. Functional redundancy between BMP antagonists is a characteristic feature observed in fish, frog, and mouse embryos (Bachiller *et al.*, 2000; Blitz *et al.*, 2003; Dal-Pra *et al.*, 2006; Khokha *et al.*, 2005). Besides providing functional compensation, negative feedback in BMP signaling is important to expand the dynamic BMP signaling range essential for proper embryonic patterning and reduce inter-individual phenotypic and molecular variability in *Xenopus* embryos (Paulsen *et al.*, 2011).

Also, the type I and type II TGF- β family receptors constantly internalize and recycle to the plasma membrane, thus resulting in a pool at the plasma membrane and an internalized pool. Rather than inducing internalization, are Rspo2/3 more likely just diverting more of the pool away from recycling and towards the lysosome?

We now performed receptor internalization assays in H1581 cells using cleavable sulfo-NHS-SS-Biotin for BMPR1A/ALK3-HA upon RSPO2 stimulation (**New Supplementary Figure 10a**). In control cells, BMPR1A/ALK3 was barely internalized, while RSPO2 triggered BMPR1A/ALK3 internalization within 20 min. This was also seen by IF *in vivo*, where 20 min treatment of *Xenopus* Rspo2 induces vesicular Bmpr1a/Alk3 staining (**New Supplementary Figure 10b-c**). Conversely, siRSPO2 treatment in H1581 cells inhibits BMPR1A/ALK3 localization to early endosome marker EEA1 (**New Figure 8p-q**). The most straightforward interpretation is therefore that RSPO2 induces BMPR1A/ALK3 endocytosis and removal from the membrane. This is now described in the model (**New Supplementary Figure 10d**).

Comment 3. *Why is only ALK3 regulated this way? Which is the most dominant BMP type I receptor in Xenopus embryos? Is it ALK3, or ALK2 or ALK6?*

As we had described in the MS, RSPO2 and -3 only bind to BMPR1A/ALK3 but not to ACVR1/ALK2 and BMPR1B/ALK6, thereby specifically regulate BMPR1A/ALK3. It will be interesting in the future to analyze the discriminatory protein motifs in the various receptor ECDs. In *Xenopus* embryogenesis, both *Bmpr1a/Alk3* and *Bmpr1b/Alk6* are expressed and they act partially redundant (Leibovich *et al.*, 2017; Schille *et al.*, 2016) (now added in the Discussion).

Comment 4. *What happens upon ligand stimulation? Do RSPO2/3 compete with BMP4? Which has the higher affinity?*

We now analyzed whether BMP4 affects RSPO2-BMPR1A/ALK3 interaction. *In vitro* binding assays show that BMP4 disrupts RSPO2-BMPR1A/ALK3 binding by around 45% (**Referee Figure 1**). We further analyzed whether BMP4 affects ZNRF3-RSPO2-BMPR1A/ALK3 ternary complex formation *in vitro* (**Referee Figure 2**). Interestingly, different from binary RSPO2-BMPR1A/ALK3 interaction, BMP4 does not disassemble the ternary ZNRF3-RSPO2-BMPR1A/ALK3 complex. Therefore, we speculate that ZNRF3-RSPO2 interaction is important to prevent BMP4 from disrupting RSPO2-BMPR1A/ALK3 binding, but this will need to be analyzed in future studies.

Comment 5. *For both the MOs and also for the transient CRISPR experiments, it is important to have more controls for off target effects. In both cases, rescues would be the best way to do this.*

We have validated our newly designed Mos and guide RNAs using rescue assays. Ventralized defects in *chordin* and *noggin* Crispants were rescued by co-injection of *chordin* and *noggin* DNA, respectively (**New Supplementary Figure 3f-i**). In addition, defects in *rspo2*^{ΔTSP} Morphants and *znrf3* Morphants were also rescued by *rspo2* mRNA and human *ZNRF3* mRNA injection, respectively (**New Supplementary Figure 6a-b; New Supplementary Figure 7d-e**).

Comment 6. *In Figure 2b, the authors show that Rspo2 can rescue defects due to Bmp4 overexpression. The authors need to show what happens when Rspo2 is expressed alone.*

Overexpression phenotypes of *rspo2*, *rspo2*^{AFU1/2} and *rspo2*^{ΔTSP} are now displayed in **New Supplementary Figure 2c**. Within the dosage which used for rescue assay in Figure 2, neither wildtype *rspo2* nor *rspo2* mutants shows any defect. We note that higher doses of *rspo2* mRNA induces gastrulation defect including bent axis and small head as reported previously (Kazanskaya *et al.*, 2004).

Comment 7. *The authors should explain in the text how the binding assays in Figure 3c and 3e were done. The text is rather cryptic on this point.*

We apologize for the brief information. We now have added scheme of experiments with the sufficient explanation in the text (**New Figure 4e and 4d**)

Comment 8. *The structure of the manuscript is a little strange in that reagents such as the R1-TSPR2 are used in earlier figures (Figure 3) and then explained later (with the data for Figure 5). The same is true for the *rspo2*ΔTSP MO. It would be better to either explain them both when they are first used (Figure 3), or wait until the figure 5, but present all the relevant data using them in the same place.*

We now restructured the ms. The data for the R1-TSP1^{R2} from previous Figure 5 is now first introduced and explained in **Figure 4i-k**, followed by further binding analysis in **New Supplementary Figure 5f** (previous Figure 3) for better flow. *rspo2*^{ΔTSP} Mo is first introduced in **Figure 5**, and the panels from previous Figure 3 are now displayed in **Figure 6f-i**.

Comment 9. *In the surface biotinylation assays, why does knockdown of ZNRF3/RNF43 not have much effect?*

We repeated the surface biotinylation assays with longer siRNA treatment, and improved the data, now showing robust increase of cell surface BMPR1A/ALK3 levels upon knockdown of ZNRF3/RNF43 (**Figure 8n**).

References

- Bachiller, D., Klingensmith, J., Kemp, C., Belo, J.A., Anderson, R.M., May, S.R., McMahon, J.A., McMahon, A.P., Harland, R.M., Rossant, J., *et al.* (2000). The organizer factors Chordin and Noggin are required for mouse forebrain development. **Nature** *403*, 658-661.
- Blitz, I.L., Cho, K.W., and Chang, C. (2003). Twisted gastrulation loss-of-function analyses support its role as a BMP inhibitor during early *Xenopus* embryogenesis. **Development** *130*, 4975-4988.
- Dal-Pra, S., Furthauer, M., Van-Celst, J., Thisse, B., and Thisse, C. (2006). Noggin1 and Follistatin-like2 function redundantly to Chordin to antagonize BMP activity. **Dev Biol** *298*, 514-526.
- de La Coste, A., Romagnolo, B., Billuart, P., Renard, C.A., Buendia, M.A., Soubrane, O., Fabre, M., Chelly, J., Beldjord, C., Kahn, A., *et al.* (1998). Somatic mutations of the beta-catenin gene are frequent in mouse and human hepatocellular carcinomas. **Proc Natl Acad Sci U S A** *95*, 8847-8851.
- Kazanskaya, O., Glinka, A., del Barco Barrantes, I., Stannek, P., Niehrs, C., and Wu, W. (2004). R-Spondin2 is a secreted activator of Wnt/beta-catenin signaling and is required for *Xenopus* myogenesis. **Dev Cell** *7*, 525-534.
- Khokha, M.K., Yeh, J., Grammer, T.C., and Harland, R.M. (2005). Depletion of three BMP antagonists from Spemann's organizer leads to a catastrophic loss of dorsal structures. **Dev Cell** *8*, 401-411.
- Leibovich, A., Steinbeisser, H., and Fainsod, A. (2017). Expression of the ALK1 family of type I BMP/ADMP receptors during gastrula stages in *Xenopus* embryos. **Int J Dev Biol** *61*, 465-470.
- Paulsen, M., Legewie, S., Eils, R., Karaulanov, E., and Niehrs, C. (2011). Negative feedback in the bone morphogenetic protein 4 (BMP4) synexpression group governs its dynamic signaling range and canalizes development. **Proc Natl Acad Sci U S A** *108*, 10202-10207.
- Schille, C., Heller, J., and Schambony, A. (2016). Differential requirement of bone morphogenetic protein receptors Ia (ALK3) and Ib (ALK6) in early embryonic patterning and neural crest development. **BMC Dev Biol** *16*, 1.

Referee Figures

Referee Figure 1. BMP4 inhibits RSPO2-BMPR1A/ALK3 binding.

For *in vitro* binding assay to analyze RSPO2-BMPR1A/ALK3^{ECD} interaction upon BMP4, RSPO2 protein was used as a bait, and BMPR1A/ALK3^{ECD}-AP was treated with or without BMP4 protein. BMPR1A^{ECD}-AP bound to RSPO2 was detected by chromogenic AP assay. Data are experimental replicates and displayed as mean \pm SD. ***P < 0.001, ****P < 0.0001 from unpaired t-test.

Referee Figure 2. BMP4 does not affect ZNRF3-RSPO2-BMPR1A/ALK3 binding.

For *in vitro* binding assay to analyze whether BMP4 affects ZNRF3-RSPO2-BMPR1A/ALK3^{ECD} interaction, ZNRF3-Fc protein was used as a bait, and BMPR1A/ALK3^{ECD}-AP was treated with or without RSPO2 and BMP4 proteins as indicated. BMPR1A/ALK3^{ECD}-AP bound to ZNRF3 was detected by chromogenic AP assay. Data are experimental replicates and displayed as mean \pm SD. ns, not significant and **P < 0.01 from unpaired t-test.

REVIEWERS' COMMENTS

Reviewer #1 (Remarks to the Author):

The authors have done an extensive and compelling rebuttal with numerous new experiments which have addressed most of our minor questions and all our major concerns. They should be commended for their sustained effort which achieve fruition in record time.

One minor point in our opinion remains with regards to our comment 24. While we appreciate that the authors performed rescue experiments showing the specificity of the *rspo2*ΔTSP morpholino, it still remains to be proven whether this particular splicing morpholino targeting the exon 4 of *Rspo2* actually leads to a stable and perfectly spliced endogenous *rspo2*ΔTSP mRNA.

Lastly and this is a friendly remark, I don't think that I am "confused" about what anti-BMP molecule is expected to do on a *Xenopus* embryo. I now fully subscribe to the authors explanation as to why *RSPO2* O/E should not overtly dorsalize the embryo. Notwithstanding it is important to clearly convey this difference to external readers who are not fully attuned to the palette of assays available to amphibian biologists. If two of the three (frog) reviewers asked the same question, maybe it reflects a legitimate interrogation ;) and not a criticism of the work.

Altogether, the manuscript is significantly improved and is now suitable for publication in *Nature* communications. I am persuaded that it will reverberate far and ignite many follow-up studies.

Reviewer #3 (Remarks to the Author):

The authors have considerably improved their manuscript with the revisions and they have adequately answered all my questions and comments. I have no further questions.

Response to the reviewers' comments

Reviewer #1 (Remarks to the Author):

The authors have done an extensive and compelling rebuttal with numerous new experiments which have addressed most of our minor questions and all our major concerns.

They should be commended for their sustained effort which achieve fruition in record time.

Thank you.

*One minor point in our opinion remains with regards to our comment 24. While we appreciate that the authors performed rescue experiments showing the specificity of the *rspo2*ΔTSP morpholino, it still remains to be proven whether this particular splicing morpholino targeting the exon 4 of *Rspo2* actually leads to a stable and perfectly spliced endogenous *rspo2*ΔTSP mRNA.*

The evidence for the splicing MO eliciting a specific TSP effect is very strong:

- *rspo2*ΔTSP Mo injection results in a BMP-specific phenotype
- it has no effect on WNT signaling unlike regular *rspo2* Mo
- *rspo2* mRNA efficiently rescues the *rspo2*ΔTSP Mo defect, ruling out off-target effects

*Lastly and this is a friendly remark, I don't think that I am "confused" about what anti-BMP molecule is expected to do on a *Xenopus* embryo. I now fully subscribe to the authors explanation as to why *RSPO2* O/E should not overtly dorsalize the embryo. Notwithstanding it is important to clearly convey this difference to external readers who are not fully attuned to the palette of assays available to amphibian biologists. If two of the three (frog) reviewers asked the same question, maybe it reflects a legitimate interrogation ;) and not a criticism of the work.*

Thanks. We have now clearly spelled-out the difference between chordin/noggin and *Rspo2* in the Discussion:

*Yet, overexpression of *rspo2* unlike of *noggin* and *chordin*, does not strongly dorsalize early embryos. The reason is that instead of sequestering BMP ligands, *RSPO2* specifically targets the BMP receptor *BMPR1A* in early *Xenopus* embryos and that *BMPR1A* and *BMPR1B* play overlapping roles in dorsoventral patterning^{50, 51}. Also unlike *noggin* and *chordin*, *rspo2* is not expressed in the organizer but is a negative feedback inhibitor of the BMP4 synexpression group, similar to the BMP pseudoreceptor *bambi* ^{44, 52}*

*Altogether, the manuscript is significantly improved and is now suitable for publication in *Nature communications*. I am persuaded that it will reverberate far and ignite many follow-up studies.*

Reviewer #2(Remarks to the Author): No comments

Reviewer #3 (Remarks to the Author):

The authors have considerably improved their manuscript with the revisions and they have adequately answered all my questions and comments. I have no further questions.

Thank you.